# Proteome-wide systems genetics identifies UFMylation as a regulator of skeletal muscle function

Jeffrey Molendijk[1,2], Ronnie Blazev[1,2], Richard J Mills[3], Yaan-Kit Ng[1,2], Kevin I Watt[1,2†], Daryn Chau[4], Paul Gregorevic[1,2], Peter J Crouch[5], James BW Hilton[5], Leszek Lisowski[6,7], Peixiang Zhang[8], Karen Reue[8], Aldons J Lusis[8,9], James E Hudson[3], David E James[10], Marcus M Seldin[4], Benjamin L Parker[1,2]*

[1]Department of Anatomy and Physiology, University of Melbourne, Melbourne, Australia; [2]Centre for Muscle Research, University of Melbourne, Melbourne, Australia; [3]QIMR Berghofer Medical Research Institute, Brisbane, Australia; [4]Department of Biological Chemistry and Center for Epigenetics and Metabolism, University of California, Irvine, Irvine, United States; [5]Department of Biochemistry and Pharmacology, University of Melbourne, Melbourne, Australia; [6]Children's Medical Research Institute, University of Sydney, Sydney, Australia; [7]Military Institute of Medicine, Warszawa, Poland; [8]Department of Human Genetics/Medicine, David Geffen School of Medicine, University of California, Los Angeles, Los Angeles, United States; [9]Department of Microbiology, Immunology and Molecular Genetics, University of California, Los Angeles, Los Angeles, United States; [10]Charles Perkins Centre, School of Life and Environmental Science, School of Medical Science, University of Sydney, Sydney, Australia

*For correspondence:
ben.parker@unimelb.edu.au

Present address: †Novo Nordisk Foundation Centre for Stem Cell Medicine, Murdoch Children's Research Institute, Parkville, Australia

**Abstract** Improving muscle function has great potential to improve the quality of life. To identify novel regulators of skeletal muscle metabolism and function, we performed a proteomic analysis of gastrocnemius muscle from 73 genetically distinct inbred mouse strains, and integrated the data with previously acquired genomics and >300 molecular/phenotypic traits via quantitative trait loci mapping and correlation network analysis. These data identified thousands of associations between protein abundance and phenotypes and can be accessed online (https://muscle.coffeeprot.com/) to identify regulators of muscle function. We used this resource to prioritize targets for a functional genomic screen in human bioengineered skeletal muscle. This identified several negative regulators of muscle function including UFC1, an E2 ligase for protein UFMylation. We show UFMylation is up-regulated in a mouse model of amyotrophic lateral sclerosis, a disease that involves muscle atrophy. Furthermore, in vivo knockdown of UFMylation increased contraction force, implicating its role as a negative regulator of skeletal muscle function.

## Editor's evaluation

This manuscript will be of broad interest to those working in the genetics of complex diseases, with the results strongly supporting the author's primary claims. Overall, this is an important study that demonstrates the power of proteomics-based systems genetics studies in the mouse.

## Introduction

Building and maintaining healthy skeletal muscles is crucial for all stages of life. Skeletal muscle makes up 30–40% of an adult human's body mass and is vital not just for breathing and movement, but also for metabolism and longevity. The maintenance of muscle function is one of the best predictors for overall health, with sarcopenia being the major contributor of age-associated frailty (*McGregor et al., 2014*). Identifying factors that regulate skeletal muscle function has great potential to improve the quality of life for humans and animals.

Systems genetics is a population-based approach that links genetic variation to complex traits (*Baliga et al., 2017*). These forward genetics approaches integrate genomics and other multi-omic data to phenotypic traits using various modelling approaches including genetic mapping and correlation analysis. Advances in high-throughput proteomic techniques have enabled the analysis of genetically diverse populations, and integration of these data with systems genetics has emerged as a powerful approach to identify novel associations between protein abundance and complex phenotypes (for a review, see *Molendijk and Parker, 2021a*). Recent large-scale plasma proteomic studies in human populations have begun to unravel complex genetic variations and their contribution to proteome diversity and disease-relevant phenotypes (*Suhre et al., 2017*; *Benson et al., 2018*; *Sun et al., 2018*; *Emilsson et al., 2018*). However, unlike studies in humans, the use of genetic reference panels (GRPs) enables accurate control of the environment, breeding patterns, and access to a range of tissues for molecular analysis. Systems genetic analyses incorporating transcriptomics, lipidomics, and/or metabolomics in GRPs have led to the discovery of a range of novel regulators of complex phenotypes ranging from insulin resistance (*Parks et al., 2015*), insulin secretion (*Keller et al., 2019*), atherosclerosis (*Bennett et al., 2015*), lipid metabolism (*Jha et al., 2018a*; *Jha et al., 2018b*; *Linke et al., 2020*), cardiac hypertrophy (*Rau et al., 2015*), cardiac diastolic dysfunction (*Cao et al., 2022*), and many more. The inclusion of proteomics into systems genetic analysis provides information on an important biological layer and has been performed in a range of GRPs including yeast (*Foss et al., 2007*; *Picotti et al., 2013*; *Parts et al., 2014*), worms (*Singh et al., 2016*), fruit fly (*Okada et al., 2016*), plants such as maize (*Hu et al., 2017*; *Jiang et al., 2019*), and several livestock such as cattle (*Boudon et al., 2020*) and pig (*Bovo et al., 2018*). The use of proteomics to analyse the liver proteome of mouse GRPs has also gained popularity and been used to analyse the BxD panel (*Wu et al., 2014*; *Williams et al., 2016*), the Hybrid Mouse Diversity Panel (HMDP) (*Ghazalpour et al., 2011*; *Parker et al., 2019*), and cohorts of the Collaborative Cross/Diversity Outbred (CC/DO) (*Chick et al., 2016*). More recently, several studies have performed proteomic analysis of additional tissues from cohorts of the BXD (*Williams et al., 2018*) and CC/DO (*Xiao et al., 2022*) and include further phenotypic associations.

Functional screening of genetic perturbations with high-throughput phenotypic measurements have identified novel regulators of muscle biology and disease. These include forward genetic mutagenesis screens to identify regulators of skeletal muscle development and locomotion in zebrafish (*Birely et al., 2005*; *Horstick et al., 2013*; *Johnson et al., 2013*; *Bennett et al., 2018*) and worms (*Beron et al., 2015*), RNAi screening of muscle size and function in fruit fly (*Kao et al., 2021*; *Graca et al., 2021*), and CRISPR/Cas9 screening of muscle cells to identify regulators of myogenesis and cell survival in dystrophy models (*Bi et al., 2017*; *Lek et al., 2020*; *Ashoti et al., 2022*). Here, we combined forward genetics via proteomic analysis of a diverse mouse panel with a targeted reverse genetics screen via AAV6 vector-mediated expression of shRNAs to knock down specific genes in bioengineered skeletal muscle to identify candidate regulators of skeletal muscle function. Our approach identified UFMylation as a regulator of skeletal muscle function that was validated in vivo.

## Results

### Genetic regulation of the mouse skeletal muscle proteome

To begin to understand how variations in the genome drive changes in the skeletal muscle proteome, we performed a proteomic analysis of gastrocnemius muscle from 73 inbred mouse strains of the HMDP that were fed a chow diet and housed under identical environmental conditions (n=2–4; 161 mice). The proteomic data were integrated with previously acquired genomic and various molecular/phenotypic data via systems genetics analysis (*Figure 1A*). Proteomics was performed with eighteen 10-plex tandem mass tag (TMT) experiments each consisting of nine strains plus a pooled common internal

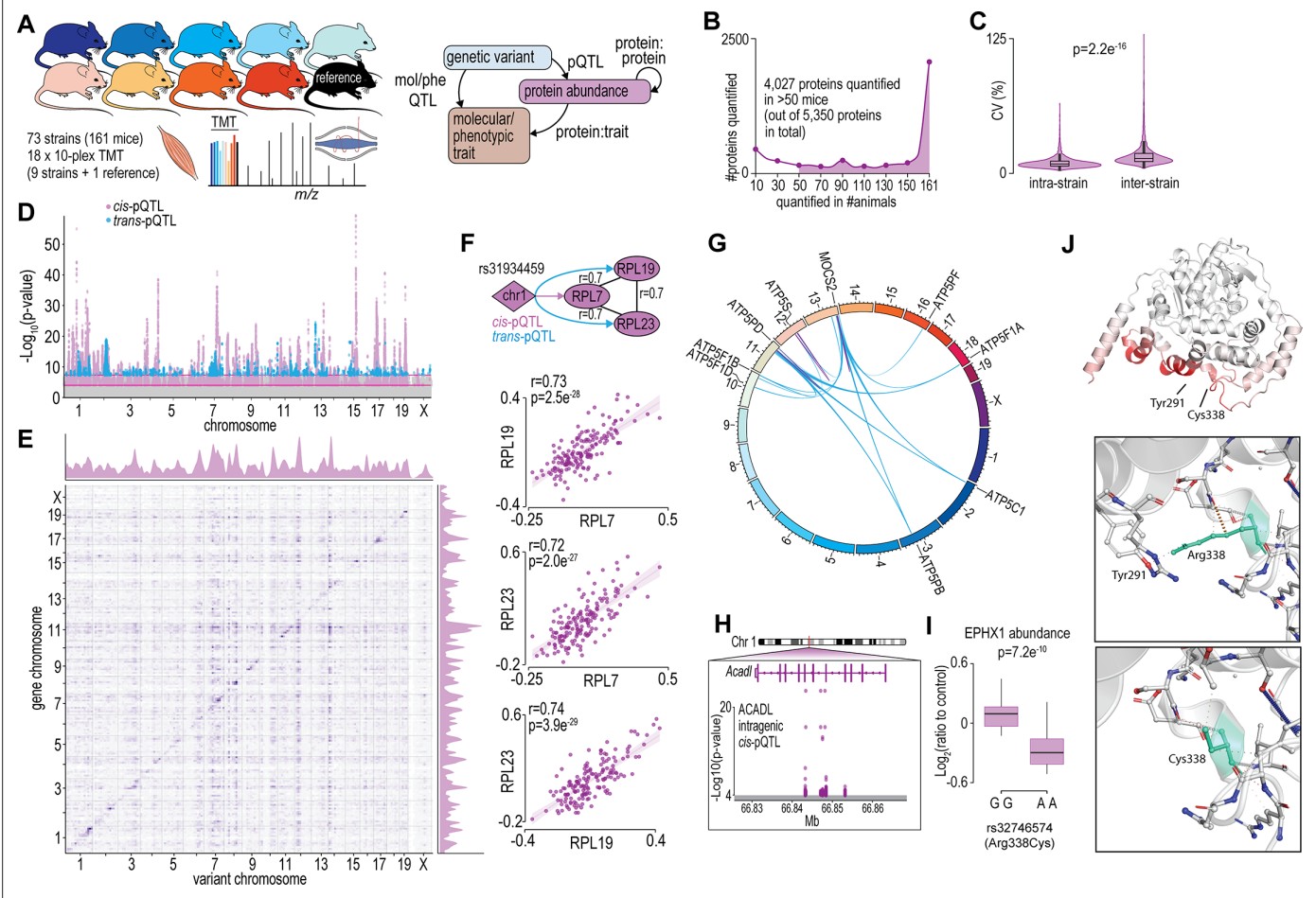

**Figure 1.** Proteome-wide systems genetics analysis of the mouse skeletal muscle proteome. (**A**) Overview of the experimental design. (**B**) Number of proteins identified. (**C**) Intra- and inter-strain coefficient of variation. (**D**) Protein-quantitative trait loci (pQTL) Manhattan plot. (**E**) pQTL variant and gene location density. (**F**) Ribosomal proteins correlation and variant network (upper), and scatterplots expressed as Log2(ratio to control) showing correlation coefficient calculated using biweight midcorrelation (n=161) (lower). (**G**) Genetic associations of variant hotspot on chromosome 13 associated with mitochondrial complex V subunits in trans. (**H**) Intragenic variants associated to ACADL abundance. (**I**) Boxplot showing variant allele associated to EPHX1 abundance (Student's t-test). (**J**) EPHX1 Arg338Cys mutation DynaMut protein flexibility analysis.

The online version of this article includes the following figure supplement(s) for figure 1:

**Figure supplement 1.** Hybrid Mouse Diversity Panel (HMDP) mouse sample dendrogram.

**Figure supplement 2.** EPHX1 Arg338Cys mutation analysis.

reference. Peptides were analysed by 2D-liquid chromatography coupled to tandem mass spectrometry (2D-LC-MS/MS) resulting in the quantification of 5350 proteins with 4027 quantified in >50 mice and 2069 proteins quantified in all 161 animals (*Figure 1B* and *Supplementary file 1*). Biological replicates showed similar proteomes, as evidenced by the hierarchical clustering dendrogram, where mice of the same strains are typically neighbouring or located closely (*Figure 1—figure supplement 1*). Median intra-strain coefficient of variation was 9.2% while inter-strain coefficient was significantly larger, suggesting reproducible quantification and genetically driven variation in the proteome was captured in the data (*Figure 1C*).

To identify possible genetic factors regulating protein abundance, we next associated single nucleotide polymorphisms (SNPs) to the abundance of 4027 skeletal muscle proteins quantified in >50 mice via a protein-quantitative trait loci (pQTL) mapping. We identified significant *cis*-regulation of the proteome with local SNPs associated to the abundance of 527 unique proteins (±10 Mb of gene; local adjusted p<1 × 10⁻⁴) and *trans*-regulation with distant SNPs (>10 Mb of the gene or on a different chromosome) associated to the abundance of 170 unique proteins (global adjusted p<5 ×

$10^{-8}$) (*Figure 1D* and *Supplementary file 2*). Visualizing SNP densities identified regions with a high number of variants (*Figure 1E*). Of particular interest are the genetic 'hotspots' (observed as vertical 'streaks' on chromosomes 7, 10, 12, and 14) abundant in *trans*-associations affecting genes in various locations. These regions agree with those observed in eQTLs of the HMDP (*Lusis et al., 2016*). We further performed variant effect predictions and annotated the position of variants relative to gene location (within a gene, i.e., intragenic or between genes, i.e., intergenic SNPs) to investigate possible mechanisms regulating protein abundance (*McLaren et al., 2016*). We observed a complex array of pQTLs on chromosome 1 with 46 and 21 proteins regulated in -*cis* and -*trans*, respectively. The majority of these pQTLs are in the distal region of the chromosome previously described as the QTL-rich region on chromosome 1 (Qrr1) with separate linkage disequilibrium (LD) blocks within 1qH2.1, 1qH5, and 1qH3 (*Mozhui et al., 2008*). The various loci in Qrr1 containing these pQTLs have been associated with a range of neural, behavioural, and cardiometabolic phenotypes, and form complex co-regulatory networks modulating a range of pathways such as RNA metabolism and translation. Here, we show a *cis*-pQTL on chromosome 1 (lead SNP rs31934459) is associated to the abundance of RPL7 and is also a *trans*-pQTL for RPL19 and RPL23. These proteins are correlated in abundance and form a co-regulated network with other 60S ribosomal proteins suggesting genetic variants regulating RPL7 abundance subsequently regulate protein complex stability/assembly (*Figure 1F* and *Supplementary file 3*). The distal region of chromosome 13 also contains a complex series of *cis*- and *trans*-pQTLs giving rise to 'hotspot' co-regulation. A *cis*-pQTL associated with MOCS2 is also a *trans*-pQTL for eight proteins all members of mitochondrial complex V (*Figure 1G*). MOCS2 is involved in molybdopterin biosynthesis and it is unclear how MOCS2 might regulate mitochondrial complex V abundance. Patients with mutations in *MOCS1* that result in mild molybdenum cofactor deficiency also display reduced mitochondrial respiration suggesting a link between molybdopterin biosynthesis and ATP production (*Grings et al., 2019*). Among the 527 proteins with a *cis*-pQTL association, 212 had an intragenic association. For example, non-coding SNPs in the fifth and sixth intron of *Acadl* were *cis*-pQTLs for ACADL (*Figure 1H*). We also identified 14 missense variants as pQTLs such as the rs32746574 variant (GG >AA; R338C), which was associated with significantly lower abundance of EPHX1 (*Figure 1I*). The R338C mutation was found to be deleterious (PROVEAN score: –5.6) (*Choi and Chan, 2015*), destabilizing (FoldX:+1.1 ΔG) (*Delgado et al., 2019*) and possibly damaging (Poly-Phen2 HumDiv: 0.62) (*Adzhubei et al., 2013*; *Figure 1—figure supplement 2*). R338C increases the flexibility of EPHX1 due to the missing interaction (hydrogen bond) between R338 and Y291 (*Figure 1J*). Aligning human and mouse EPHX1 protein structures revealed that the missense mutation causing R338C (yellow) is in close proximity to the catalytic (blue) and disease-related sites (red) identified by *Gautheron et al., 2021*; *Figure 1—figure supplement 2*. Taken together, our proteomic analysis of the HMDP has helped define the genetic factors potentially regulating the abundance of hundreds of skeletal muscle proteins.

## Systems genetics integration of the skeletal muscle proteome with molecular and phenotypic traits

The renewable nature of extensively characterized inbred mouse strains from genetic references panels such as the HMDP and the BxD allow for the integration of data across multiple cohorts (*Lusis et al., 2016*; *Ashbrook et al., 2021*). We focused our analysis on a subset of 300 molecular or phenotypic traits incorporating various plasma metabolites, lipids, and cytokines; whole body measurements such as glucose/insulin sensitivity and body composition/organ weights; and muscle phenotypes such as cardiac and skeletal muscle function previously quantified in the same strains of mice in the HMDP (*Lusis et al., 2016*). Note that data integration is performed at the strain level, since the proteomic data was not generated from the same mice, as those used in previous studies. *Supplementary file 4* summarizes all the phenotypic data integrated in the current study and includes data sources. Cumulatively, these previous systems-level mouse studies identified hundreds of loci associated to molecular or phenotypic traits (mol/pheQTLs) (*Figure 2A* and *Supplementary file 5*). We next devised a three-step systems genetic analyses to associate these molecular or phenotypic traits to the skeletal muscle proteome that included: (1) identification of SNPs shared between skeletal muscle *cis*-pQTLs and mol/pheQTLs, (2) correlation and supervised multivariate associations between the abundance of skeletal muscle proteins and each molecular or phenotypic trait, and (3) comparison of protein abundance and molecular or phenotypic differences between allelic variations (*Figure 2B*). All data can be browsed at

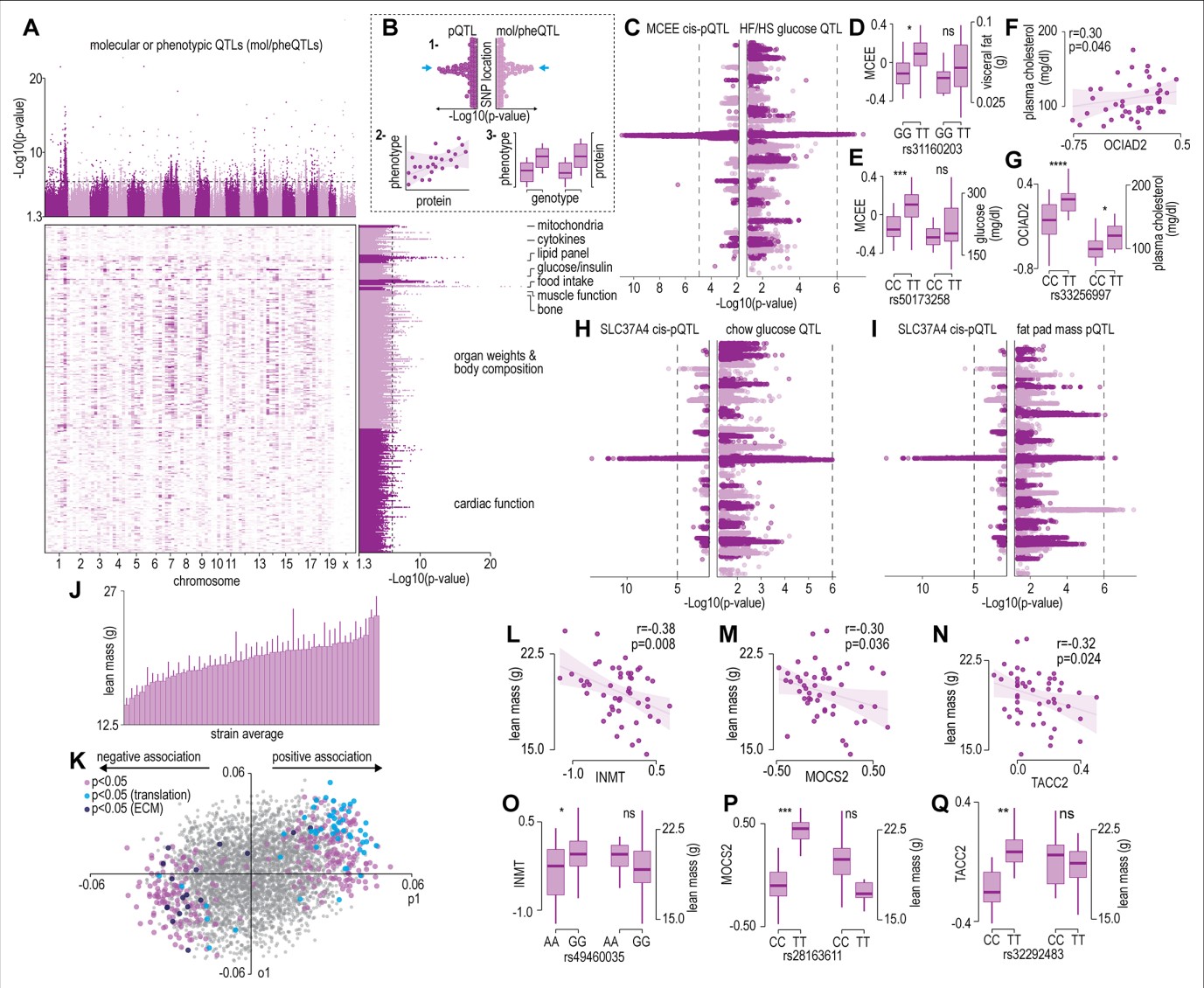

**Figure 2.** Protein and phenotype quantitative trait locus (QTL) analysis. (**A**) Manhattan plot and genomic location distribution of mol/pheQTLs. (**B**) Overview of the three-step integrative analysis approach. (**C**) Mirrored Manhattan plots of MCEE and glucose QTLs. (**D**) Allelic variant boxplots of rs31160203 for MCEE and visceral fat. (**E**) Allelic variant boxplots of rs50173258 for MCEE and glucose. (**F**) Correlation scatterplot of OCIAD2 abundance expressed as Log2(ratio to control) and plasma cholesterol concentrations. (**G**) Allelic variant boxplots of rs33256997 for OCIAD2 and plasma cholesterol. (**H**) Mirrored Manhattan plots of SLC37A4 and glucose QTLs. (**I**) Mirrored Manhattan plots of SLC37A4 and fat pas mass QTLs. (**J**) Average distribution of lean mass per mouse strain. (**K**) Orthogonal partial least-squares (OPLS) loading plot of proteins explaining the variance related to strain lean mass. Separation on the x-axis shows variation related to the predictive component (p1), whilst the y-axis shows the orthogonal component (o1). Highlighted points reflect Student's correlation p-values for multiple biweight midcorrelations of proteins correlated with lean mass (–0.3< r > 0.6, p<0.05). Correlation of lean mass and the protein abundance expressed as Log2(ratio to control) of INMT (**L**), MOCS2, (**M**) and TACC2 (**N**). Allelic variant boxplots of selected single nucleotide polymorphisms (SNPs) with lean mass and INMT (rs49460035) (**O**), MOCS2 (rs28163611) (**P**), and TACC2 (rs32292483) (**Q**). *p<0.05, **p<0.01, ***p<0.001, ****p<0.0001 by Student's t-test.

muscle.coffeeprot.com and includes querying at the protein- or phenotype-level followed by several interactive visualizations. Analysis of this new resource identified hundreds of associations allowing for the prioritization of potential causal proteins regulating molecular or phenotypic traits. We present a few examples of phenotype-protein associations below. For example, a locus on chromosome 7 contained *cis*-pQTLs associated to MCEE that were shared with fasting glucose in mice subject to high-fat/high-sugar feeding (***Figure 2C***). Homozygous allelic variation of SNPs were associated to MCEE protein abundance and also a trend for greater visceral adiposity and fasting glucose on a chow diet (***Figure 2D–E***). MCEE functions as a methylmalonyl-CoA epimerase important for propionyl-CoA

metabolism. Rare autosomal recessive missense mutations in *MCEE* have been identified in patients with methylmalonic aciduria (*Bikker et al., 2006*). We also identified *cis*-pQTLs associated to the abundance of both OCIAD1 and OCIAD2; two neighbouring genes located on chromosome 5. Intragenic SNPs in both genes were associated with several cardiometabolic parameters of adiposity including percentage body fat assessed by nuclear magnetic resonance, retroperitoneal fat mass, circulating free fatty acids, HDL, and cholesterol. Furthermore, the abundance of OCIAD1 and OCIAD2 were positively correlated to several of these traits, and homozygous allelic variation of SNPs in both the *Ociad1* and *Ociad2* loci were associated to protein abundance and cholesterol (*Figure 2F–G*). Very little is known about the functions of OCIAD1/2 but recent data have revealed a role in mitochondrial complex III assembly (*Le Vasseur et al., 2021*) and human GWAS analysis has identified variants in both the *OCIAD1* and *OCIAD2* loci are associated with susceptibility to type 2 diabetes (*Vujkovic et al., 2020*). We also identified genetic variants associated to the abundance of the ER resident glucose-6-phosphate transporter SLC37A4 that co-localize to fasting glucose and fat pad mass in mice fed a chow diet (*Figure 2H–I*). SLC37A4 has enhanced expression in the liver, gut, and kidney, and plays a role in the regulation of glycogenolysis and gluconeogenesis, however, its role in skeletal muscle metabolism has been comparatively less studied. Patients with mutations in SLC37A4 present with several metabolic complications, particularly hepatomegaly and enlarged kidneys due to the accumulation of glycogen but also often have signs of dyslipidaemia and hypoglycemia.

We next analysed associations of the skeletal muscle proteome to lean mass which showed a range of genetic variation in female mice fed a chow diet (*Figure 2J*). Using supervised multivariate and pairwise correlation analysis, we identified 300 proteins positively or negatively associated to lean

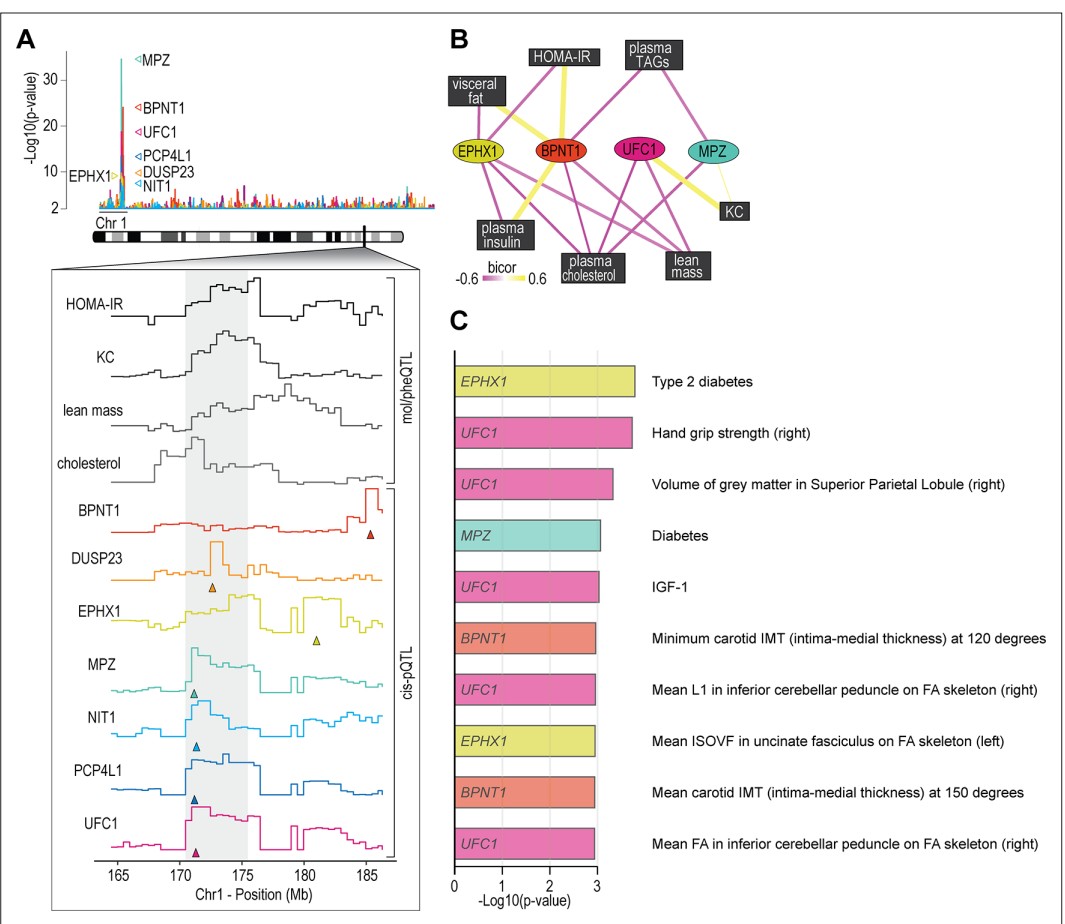

**Figure 3.** Proteome-phenotype associations of Qrr1 region on chromosome 1. (**A**) Manhattan plot of selected genes located near the Qrr1 region, with corresponding traits. (**B**) Protein-trait correlation network. (**C**) Top 10 GeneBass associations from the 'UK BioBank Assessment Centre' and 'Biological samples' categories, excluding 'Touchscreen', 'Medications', and 'Operations' categories.

mass (–0.3 < r > 0.6, p<0.05) (*Figure 2K*). Proteins involved in translation including ribosomal subunits and elongation initiation factors were positively correlated while we observed several extracellular matrix proteins including collagens and cathepsins to be negatively correlated with lean mass. Allelic variations were integrated to highlight several trends such as negative correlations between INMT, MOCS2, and TACC2 versus lean mass (*Figure 2L–N*), where alleles which reduced protein abundance (rs49460035, rs28162611, and rs32292483) generally displayed lower lean mass (*Figure 2O–Q*). We also investigated the relationships between mol/pheQTL's and skeletal muscle *cis*-pQTLs in or around the Qrr1 region, located distal on chromosome 1. *Figure 3A* displays regional association plots of pQTLs (arrows indicating gene location) shared with mol/pheQTLs including HOMA-IR, lean mass, plasma concentrations of keratinocyte-derived growth factor (KC; CXCL1), and plasma cholesterol. Two genomic regions are associated with HOMA-IR which contains *cis*-pQTLs for MPZ, NIT1, PCP4L1, and UFC1 in the first region while EPHX1 is located in the second more distal region. It is important to note that these regions were also associated with plasma insulin concentrations and as such, variations in the expression of these proteins may play greater roles in other tissues such as the pancreas. The QTL for plasma cholesterol and KC concentrations were only shared to the first region of the HOMA-IR blocks while the QTL for lean mass was distributed across both regions. To further investigate these associations, we correlated the abundance of the proteins in skeletal muscle to the molecular or phenotypic traits (*Figure 3B*). Our data identified a negative correlation between UFC1, lean mass, and plasma cholesterol. UFC1 is the E2 ligase for the post-translational modification of ubiquitin-fold modifier 1 (UFM1) to target proteins through UFMylation. We also identified negative correlations between MPZ and plasma cholesterol/triacylglycerols (TAGs). It is unclear if MPZ plays a causative role in the regulation of lipid metabolism, however diabetic peripheral neuropathy is associated with reduced expression of Mpz, myelin abnormalities, and several defects in lipid metabolism (*Cermenati et al., 2012*). Positive correlations were found between BPNT1 and plasma insulin, HOMA-IR, and visceral fat while also negatively correlated with plasma cholesterol, TAGs, and lean mass. BPNT1 is involved in phosphatidylinositol phosphate and adenosine phosphate metabolism. Mice lacking *Bpnt1* display severe liver defects but the role of this enzyme in whole body energy metabolism and insulin sensitivity is currently unknown (*Hudson et al., 2013*). Finally, we observed an overall negative correlation between the abundance of EPHX1 and HOMA-IR, visceral fat, plasma insulin, and cholesterol. Our correlation results are further corroborated by publicly available UK Biobank gene-trait associations accessed through the Genebass webserver (*Karczewski et al., 2022*; *Figure 3C*). Of particular interest are the associations between *EPHX1*/type 2 diabetes, *UFC1*/hand grip strength, and *UFC1*/IGF-1. The associations of UFC1 with grip strength (UK BioBank, human) and lean mass (HMDP, mouse) indicate a potential role of this protein in skeletal muscle compositions and/or functional capacity.

## Targeted functional genomic screen in bioengineered skeletal micro-muscles

To validate potential causal regulators of muscle function, we targeted genes encoding novel skeletal muscle pQTLs and molecular/phenotypic associations and performed a targeted functional genomic screen in human skeletal micro-muscles (hµMs) (*Mills et al., 2019*). We focused on proteins with negative associations to lean mass, grip strength, or other metabolic traits, and generated a total of 27 individual recombinant AAV serotype 6 viral vectors expressing shRNA (rAAV6:shRNAs) to knock down the expression of these proteins in an arrayed fashion. hµMs were grown around flexible pillars to assess contractile force during electrical stimulation, and transduced following differentiation and maturation to limit effects on the myogenic program (*Figure 4A*). Electrical stimulation was applied to induce either a high-frequency tetanic contraction for assessment of maximum force producing capacity or stimulated with sustained lower frequency for assessment of endurance/fatigue. Following this protocol, hµMs were analysed by proteomics which quantified 17/27 targets with 13 targets significantly reduced in abundance by rAAV6:shRNA (*Figure 4B*). Knockdown of UFC1, MCEE, TOM1L2, and SH3BGR was confirmed at the protein level and resulted in significant increases in maximum force production during the tetanic contractions (*Figure 4C*). These effects of knockdown were consistent with the observed negative correlation between lean mass and the abundance of UFC1, MCEE, and SH3BGR in skeletal muscle of the HMDP providing evidence for a causal regulation of muscle function. The fatigue protocol resulted in 20% decline in muscle function in control scramble-treated

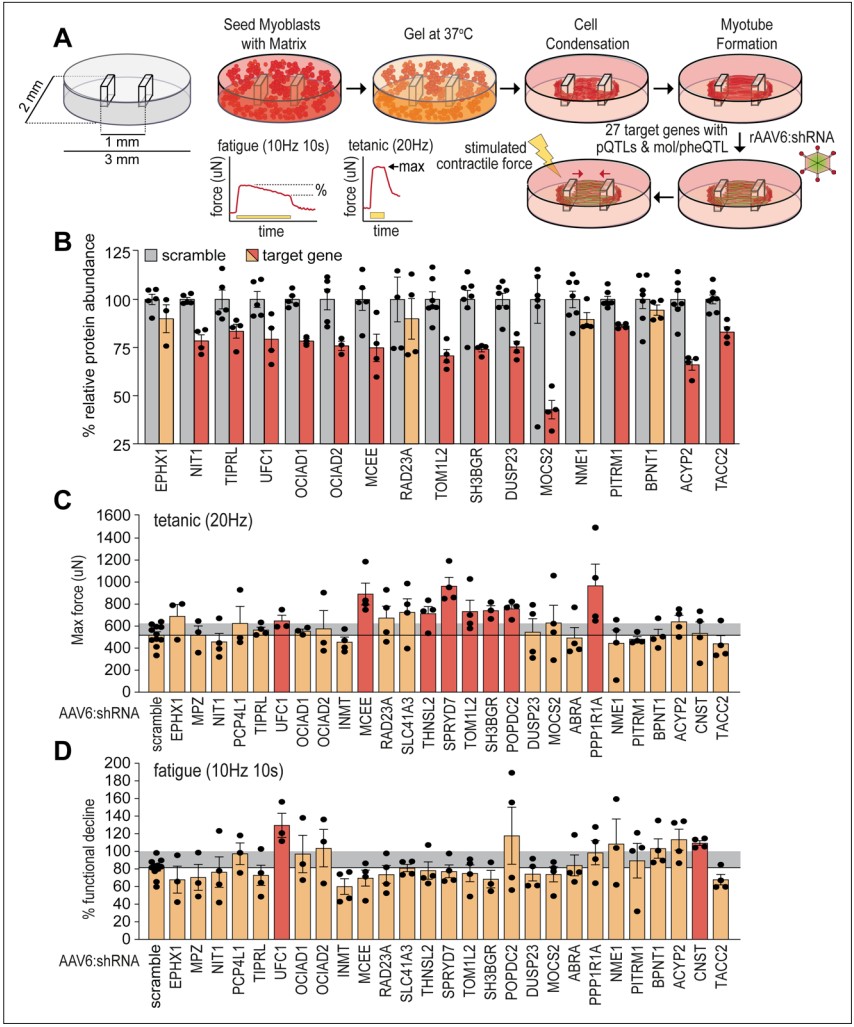

**Figure 4.** Functional screening of skeletal muscle function. (**A**) Overview of experimental design. (**B**) Knockdown efficiency of target proteins (n=4–10). (**C**) Maximum tetanic force, and (**D**) % fatigue of rAAV6:shScramble and target proteins. Red: q<0.05; yellow: q>0.05 (Student's t-test relative to scramble with Benjamini-Hochberg FDR).

hμMs (*Figure 4D*). Remarkably, knockdown of UFC1 increased force production during the fatigue protocol. Knockdown of CNST also protected against fatigue, but we were unable to identify CNST in the proteomics analyses to confirm knockdown. Taken together, these data identify potential causal negative regulators of muscle function.

## UFMylation regulates skeletal muscle function

Our data suggest that the regulation of UFC1 and subsequent changes in UFMylation may play a key role in muscle function. We first investigated the regulation of UFMylation in a mouse model of amyotrophic lateral sclerosis (ALS), a rapidly progressive adult-onset disease that involves substantial muscle atrophy (*Pansarasa et al., 2014*). The model involves progressive muscle atrophy from 11 weeks of age due to transgenic expression of an ALS-causing mutation in superoxide dismutase 1 (SOD1(G37R)) (*Wong et al., 1995*). At 25 weeks of age, we observed a significant increase in conjugated and free UFM1, UFC1, and UFSP2 (a deUFMylase) in gastrocnemius skeletal muscles of SOD1(G37R) mice, which was independent of any changes in the abundance of BiP chaperone as a marker of ER stress (*Figure 5A–B*). These data suggest the overall pool of UFM1 increases and there is an increase in UFMylation flux. We provide the first evidence of changes in UFMylation following muscle atrophy in vivo. We next manipulated in vivo UFMylation levels by injecting rAAV6:shRNA into tibialis anterior (TA) and extensor digitorum longus (EDL) skeletal muscles of 8-week-old C57BL/6J

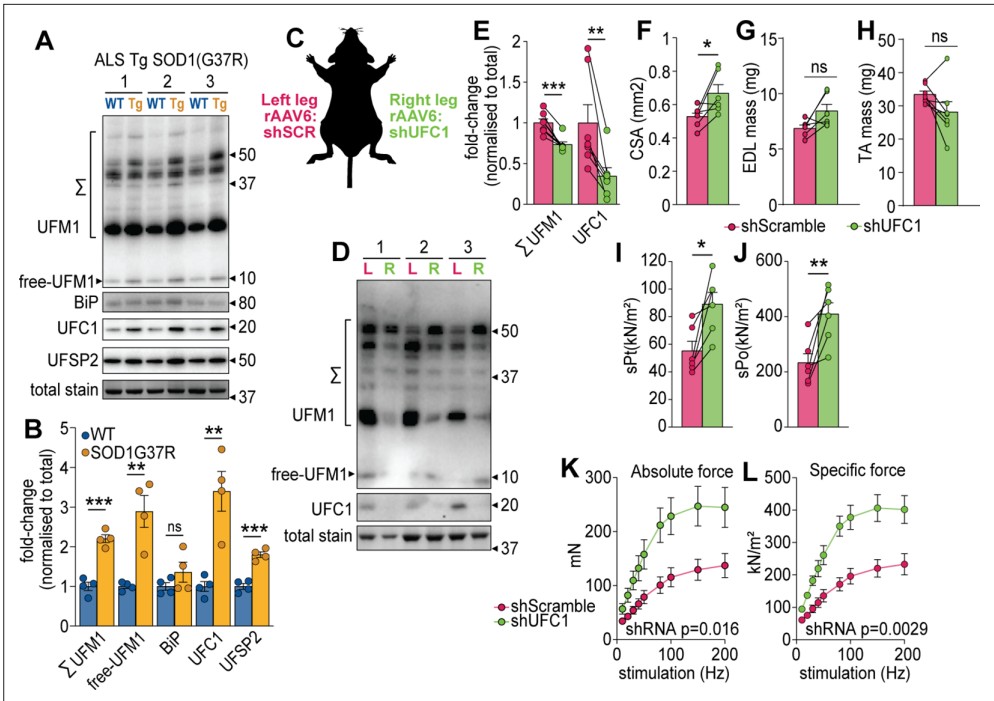

**Figure 5.** UFMylation is regulated in atrophy and influences skeletal muscle function. (**A**) Western blot and (**B**) densitometry of UFMylation and BiP chaperone in a gastrocnemius muscle of a mouse model of amyotrophic lateral sclerosis (ALS). (**C**) Overview of the experimental design. (**D**) Western blot of extensor digitorum long (EDL) muscles treated with rAAV6:shScramble (red, left leg (L)) and rAAV:shUFC1 (green, right leg (R)). (**E**) Densitometry of western blot (n=6). (**F**) Muscle cross-sectional area (CSA) (n=6). (**G**) EDL mass and (**H**) tibialis anterior (TA) mass (n=6). Ex vivo analysis of contraction force in EDL muscles showing (**I**) single twitch contraction force normalized to CSA (sPt), (**J**) tetanic contraction force normalized to CSA (sPt), and (**K**) absolute, and (**L**) specific force normalized to CSA following shUFC1 or scrambled control. *p/q-value<0.05; **p/q-value<0.01; ***p/q-value<0.005; (B–C) paired Student's t-test; (E–J) paired Student's t-test; (K–L) two-way ANOVA.

The online version of this article includes the following source data for figure 5:

**Source data 1.** Zip file containing uncropped western blot image files as Image Lab Documents, tiff files, and a summarized.pdf highlighting the lane identifications, highlighted bands used to create **Figure 5A**, antibody information, and all densitometry results for each individual sample.

**Source data 2.** Zip file containing uncropped western blot image files as Image Lab Documents, tiff files, and a summarized.pdf highlighting the lane identifications, highlighted bands used to create **Figure 5D**, antibody information, and all densitometry results for each individual sample.

mice in a paired experimental design, where muscles of the left leg received rAAV6:shScramble and muscles of the right contralateral leg received rAAV6:shUFC1 (**Figure 5C**). Following 12 weeks of transduction, we observed significant reductions in the abundance of UFC1 and conjugated UFM1 (**Figure 5D–E**). There was a subtle but significant increase in whole muscle cross-sectional area (CSA) but no change in muscle mass (**Figure 5G–H**). EDL muscles were subjected to ex vivo assessments of muscle function using electrically induced contractions. Knockdown of UFC1 increased specific force production in response to a single twitch contraction (sPt) and generated almost a doubling of peak force (sPo), normalized to CSA (**Figure 5I–J**). Furthermore, both the absolute and specific tetanic force increased across all stimulation frequencies tested following knockdown of UFC1 (**Figure 5K–L**).

We next performed a more detailed cellular and molecular analysis. First, we analysed fiber-type composition of TA muscles by immunofluorescence microscopy of which revealed no differences in the abundance of myosin isoforms but there was a trend for a decrease in the total number of muscle fibers following knockdown of UFC1 (**Figure 6A–C**). Next, we performed a proteomic analysis of EDL muscles which quantified 5909 proteins of which 573 were regulated in abundance following knockdown of UFC1 (**Figure 6D** and **Supplementary file 6**). The top up-regulated pathways included proteins associated with translation, muscle contraction, and signal recognition particle

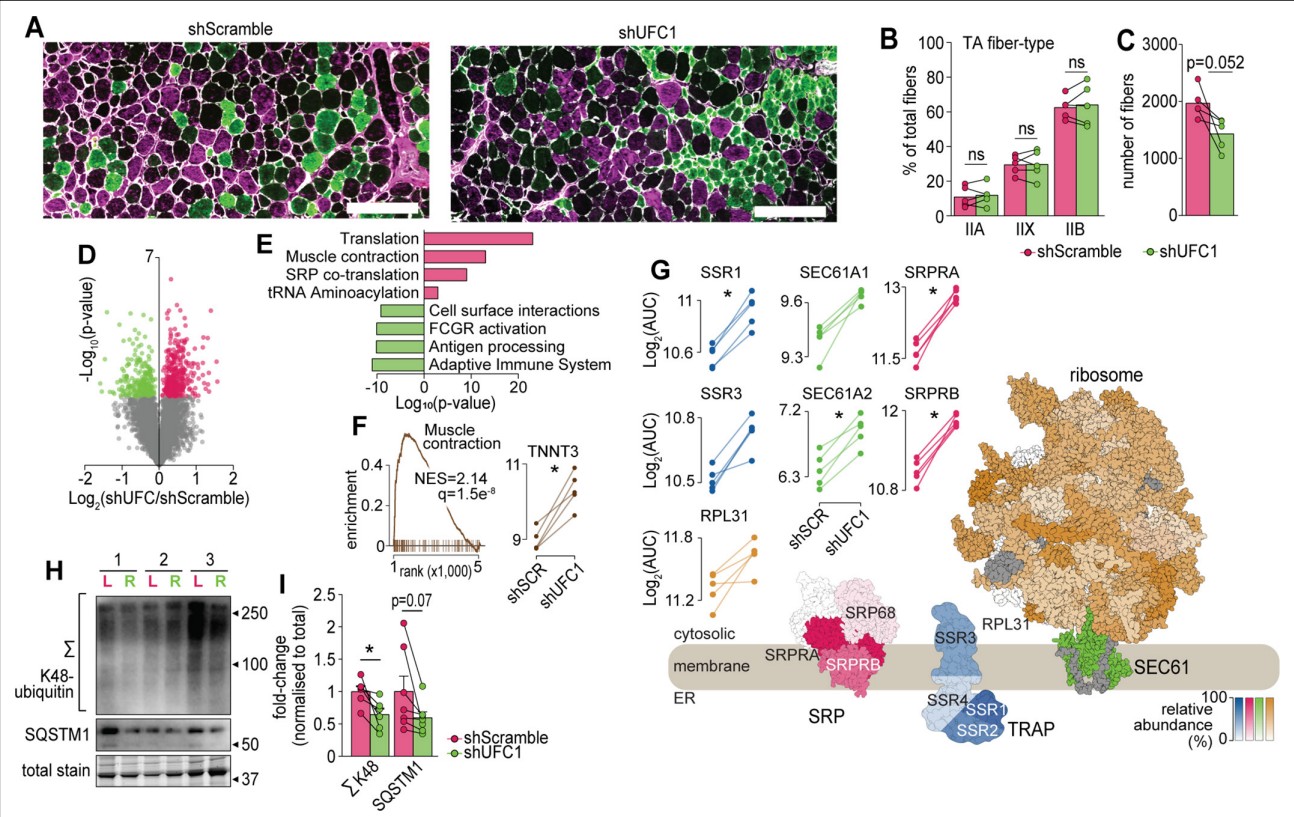

**Figure 6.** Characterization of skeletal muscles following UFC1 knockdown. (**A**) Representative immunofluorescence microscopy of fiber-type composition in tibialis anterior (TA). Myosin heavy chain isoforms (MYH2, green, type IIa; MYH4, purple, type IIb; MYH1, unstained, type IIx) while laminin is white. Scale bar = 200 μm. (**B**) TA fiber-type distribution, and (**C**) TA total fiber number (n=5). (**D**) Volcano plot and (**E**) gene set enrichment analysis of proteins affected by UFC1 knockdown. (**F**) Enrichment plot of the muscle contraction gene set (REACTOME_MUSCLE_CONTRACTION, MSigDB C2 collection) and paired analysis of TNNT3. (**G**) Knockdown of UFC1 up-regulates the ribosome-SEC61 complex, signal recognition particle, and translocon-associated protein. Protein constituents of each structure were coloured based on the relative increased abundance following shUFC1, where the colours are scaled based on the relative fold change per complex (signal recognition particle [SRP] – red; translocon-associated protein (TRAP) – blue; SEC61 – green; ribosome – yellow/orange, grey – not measured). (**H**) Western blot of extensor digitorum longus (EDL) muscles treated with rAAV6:shScramble (red, left leg (L)) and rAAV:shUFC1 (green, right leg (R)). (**I**) Densitometry of western-blot (n=6). *p/q-value<0.05; (B, C, I): paired Student's t-test; (F–G): paired Student's t-test with Benjamini-Hochberg FDR.

The online version of this article includes the following source data for figure 6:

**Source data 1.** Zip file containing uncropped western blot image files as Image Lab Documents, tiff files, and a summarized.pdf highlighting the lane identifications, highlighted bands used to create *Figure 6H*, antibody information and all densitometry results for each individual sample.

(SRP)-mediated translocation to the ER, while down-regulation was observed in immune-related pathways and cell surface interactions (*Figure 6E*). The up-regulation of contractile proteins following knockdown of UFC1 includes fast-type Troponin T (TNNT3), which binds tropomyosin, and the protein with the largest fold change (>2.5-fold) (*Figure 6F*). The most up-regulated proteins in the translational machinery were SRPA/B of the SRP complex, SSR1/2/3/4 of the TRAP complex, and SEC61A/B (*Figure 6G*). We also observed a trend for increased RPL31 which mediates binding of SRP to the ribosome (*Pech et al., 2010*). The core subunits of the proteasome, and atrogenes including TRIM63 and ASB2 were not regulated following knockdown of UFC1 (*Supplementary file 6*). However, the proteasome activating complex PSME1/2 was down-regulated with UFC1 knockdown and western blotting revealed a decrease in K48-linked ubiquitination (*Figure 6H–I*). We also observed a complex regulation of autophagy-associated proteins such as up-regulation of GABARAP and ATG3, and down-regulation of ATG4A, WIPI1, and SQSTM1, the latter which trended to decrease by >1.8-fold with both western blotting and proteomics (*Figure 6H–I* and *Supplementary file 6*). Taken together, our data reveal that UFMylation is regulated during atrophy and plays a role in skeletal muscle function via modulating proteostasis mechanisms and contractile proteins.

# Discussion

Genetic determinants of diversity in skeletal muscle attributes remain incompletely defined, and thus an opportunity to transform our understanding of muscle biology. We performed a proteomic analysis of skeletal muscle in a genetically diverse mouse panel and integrated the data with a variety of molecular and phenotypic traits that can be browsed at https://muscle.coffeeprot.com to potentially uncover new biology relevant for human disease. We demonstrate the utility of this resource by performing a targeted functional screen in human bioengineered skeletal muscle which revealed UFMylation as a negative regulator of muscle function that was further validated in vivo. Targets of UFMylation include nuclear proteins Histone H4 (*Qin et al., 2019*), MRE11 (*Lee et al., 2021*), and ACS1 (*Yoo et al., 2014*) with the latter particularly relevant to the current study given mutations in ACS1 are associated with muscular atrophy (*Davignon et al., 2016*). However, the most abundant target of UFMylation in mammalian cell culture and validated by two independent studies is RPL26, a ribosomal-associated protein in close proximity to the interaction sites of RPL31 and the SEC61 complex (*Walczak et al., 2019*; *Wang et al., 2020*). Indeed, the UFM1 E3 ligase, UFL1, is targeted to the ER via UFBP1 (DDRGK1) (*Wang et al., 2020*), and UFMylation is required for ER-phagy (*Liang et al., 2020*) and ER-associated protein degradation (ERAD) (*Walczak et al., 2019*). Mechanistically, RPL26 UFMylation promotes the degradation of a translocation-arrested ER protein and up-regulation may play a role in enhanced secretory flux (*Wang et al., 2020*). Hence, UFMylation plays a positive role in ER stress and cellular protection, and it is potentially paradoxical that reducing UFMylation can improve muscle contractile function. One potential explanation is an acute reduction in UFMylation leads to proteome compensatory remodelling and up-regulation of translational machinery which provide later benefits to muscle function. A limitation of our approach is that the abundance of UFMylation enzymes was measured, but not the degree of UFMylation among proteins. While RPL26 was not significantly regulated following UFC1 knockdown, it is possible that the UFMylation status of RPL26 was altered. The positive enrichment of ribosomal subunits including RPL31, and SRP co-translational factors (SSR1/3, SEC61α, SRPRB) located in close proximity to RPL26 suggests an involvement of UFMylation in this process. In further support of a negative association of UFMylation and muscle function, mutations in UFSP2 have been identified which increase UFMylation and result in musculoskeletal dysplasia (*Watson et al., 2015*; *Zhang et al., 2020*). Furthermore, the expression of UFC1 is also up-regulated in skeletal myocytes differentiated from induced pluripotent stem cells derived from familial ALS (C9ORF72 mutations) (*Lynch et al., 2021*), and the expression of UBA5, the E1 ligase for UFMylation has recently been associated with Becker muscular dystrophy (*Xu et al., 2021*). Interestingly, genetic ablation of UFMylation in cell culture reduces viral-mediated interferon production and correlates with our observed down-regulation of proteins involved in the innate immune response following UFC1 knockdown suggesting an overall reduction in inflammation (*Snider et al., 2022*).

## Ideas and speculation

A major question arising from our data is how does a reduction in UFMylation lead to the up-regulation of contractile proteins? Is it driven by a reduction in K48-linked ubiquitination and potential down-regulation of protein degradation or an increase in protein synthesis despite the major translational machinery up-regulated being associated with ER targeting and SRP/SEC61, or a combination of both? Furthermore, how does the regulation of UFMylation change components of the autophagy system? It is likely that the identification of additional UFMylation substrates may further unravel these mechanisms. Collectively, our systems genetics and functional screening strategy provides a rich resource to further explore mechanisms governing skeletal muscle metabolic function, and we demonstrate its use by identifying UFMylation as an important modification for muscle biology.

## Limitations of study

We only performed proteomics on female mice from the HMDP, whereas phenotypic analyses were performed on several separate cohorts using both sexes. Clearly, sex interacts with common genetic variation to influence many outcomes. In this light, an advantage of renewable resources such as the HMDP is to allow the same genetic background to be assayed across multiple studies. Thus, as more data becomes available, which aspects of this study are either female-specific or penetrant across both sexes will be easily addressed. Another potential limitation of our study is that the associations between genetic variants and proteins/phenotypes may be either causal or reactive in nature. That

is, a genetic variant may drive a change in protein abundance which then regulates a phenotype, or the genetic variant changes the phenotype which subsequently changes the abundance of the protein. Furthermore, associations may have pleiotropic affects where a genetic variant may regulate the abundance of multiple proteins which modulate a phenotype either via horizontal or vertical pleiotropy. When interpreting genetic associations, it is also relevant to consider the overall heritability of a given trait. Specifically, broad sense heritability measures can inform the overall confidence in linking genotype to phenotype and inferring genetic interactions with environment and sex (*Andreux et al., 2012*; *Seldin et al., 2019*; *Ashbrook et al., 2021*). The genetic repeatability (R) for each trait, as determined using the rptR workflow, is reported in *Supplementary file 4* (*Stoffel et al., 2017*). For traits which exhibit a high degree of technical variability such as cardiac function or grip strength, these estimates provide a quantitative metric with which to guide genetic contributions. To further prioritize causal associations focusing on muscle function, we targeted genes containing skeletal muscle *cis*-pQTLs that were also associated to molecular or phenotypic traits and performed a knockdown screen in hμM. Here, our goal was to identify negative regulators of adult muscle function and we utilized rAAV6:shRNA vectors applied to the micro-muscles post-differentiation to model mature skeletal muscle. Hence, the results of our screen may not be relevant for the study of myogenesis and muscle regeneration. For the first time, we show that UFMylation is regulated in an atrophy model of ALS motor neuron disease and further studies are required to investigate the regulation of UFMylation in other diseases of muscle wasting such as dystrophy, cachexia, and sarcopenia. Furthermore, additional studies are warranted to investigate if the regulation of UFMylation can provide functional benefits during atrophy. It is also important to note that we used rAAV6 vectors which provides high tropism to terminally differentiated myofibers (*Blankinship et al., 2004*) and hence, additional methodologies are required to investigate the role of UFMylation in other cell types including defective neuronal cell populations.

## Materials and methods
### HMDP animals
All mice were from The Jackson Laboratory and were subsequently bred and housed at University of California, Los Angeles, to generate offspring used in this study as previously described (*Parks et al., 2015*; *Parks et al., 2013*). Only female mice were used and housed at 22°C (±1°C) on a 12 hr light/dark cycle and ad libitum access to food and water with a chow diet (Ralston Purina Company – 5001) until 8–10 weeks of age before being fasted for 14 hr in a fresh cage. Animals were anaesthetized, exsanguinated, and gastrocnemius muscles immediately removed and snap-frozen. All protocols for these studies were approved by the Institutional Care and Use Committee (IACUC) at University of California, Los Angeles. A list of mice used in this study is shown in the 'Key resources table'.

### Proteomics sample preparation
Muscle tissue from the HMDP were lysed in 6 M guanidine HCL (Sigma; #G4505), 100 mM Tris pH 8.5 containing 10 mM tris(2-carboxyethyl)phosphine (Sigma; #75259) and 40 mM 2-chloroacetamide (Sigma; #22790) by tip-probe sonication. The lysate was heated at 95°C for 5 min and centrifuged at 20,000 × *g* for 10 min at 4°C. The supernatant was diluted 1:1 with water and precipitated overnight with five volumes of acetone at –20°C. The lysate was centrifuged at 4000 × *g* for 5 min at 4°C and the protein pellet was washed with 80% acetone. The lysate was centrifuged at 4000 × *g* for 5 min at 4°C and the protein pellet was resuspended in Digestion Buffer (10% 2,2,2-trifluoroethanol [Sigma; #96924]) in 100 mM HEPES pH 7.5. Protein was quantified with BCA (Thermo Fisher Scientific) and normalized in Digestion Buffer to a final concentration of 2 μg/μl. Protein was digested with sequencing grade trypsin (Sigma; #T6567) and sequencing grade LysC (Wako; #129-02541) at a 1:50 enzyme:substrate ratio overnight at 37°C with shaking at 2000× rpm. Eight μg of peptide was directly labelled with 32 μg of 10-plex TMT (lot #QB211242) in 20 μl at a final concentration of 50% acetonitrile for 1.5 hr at room temperature. The reaction was de-acylated with a final concentration of 0.3% (w/v) hydroxylamine and quenched with a final concentration of 1% trifluoroacetic acid (TFA). Each 10-plex experiment contained nine different strains with a tenth reference label (131 isobaric label) made up of the same peptide digest from pooled mix of C57BL/6J muscles. The sample identity and labelling channels have been uploaded as a table with the.raw proteomic data to the PRIDE

ProteomeXchange (see Data availability section). Following labelling, the peptides from each of the 18 TMT 10-plex batches were pooled and purified directly by styrene divinylbenzene reversed-phase sulfonate (SDB-RPS) microcolumns, washed with 99% isopropanol containing 1% TFA and eluted with 80% acetonitrile containing 2% ammonium hydroxide followed by vacuum concentration. Peptides were resuspended in 2% acetonitrile containing 0.1% TFA and 30 µg of peptide was fractionated on an in-house fabricated 25 cm × 320 µm column packed with C18BEH particles (3 µm, Waters). Peptides were separated on a gradient of 0–30% acetonitrile containing 10 mM ammonium formate (pH 7.9) over 60 min at 6 µl/min using an Agilent 1260 HPLC and detection at 210 nm with a total of 48 fractions collected and concatenated down to 12 fractions. Skeletal muscle micro-muscles were lysed in 4% sodium deoxycholate in 100 mM Tris pH 8.5 containing 10 mM tris(2-carboxyethyl)phosphine and 40 mM 2-chloroacetamide by tip-probe sonication. The lysate was heated at 95°C for 5 min and centrifuged at 18,000 × $g$ for 10 min at 4°C. Protein was digested with 0.2 µg of sequencing grade trypsin and 0.2 µg of sequencing grade LysC overnight at 37°C. Peptides were first diluted with 100% isopropanol, mixed and then acidified with TFA to a final concentration of 50% isopropanol, 0.1% TFA. Peptides were desalted with SDB-RPS microcolumns, washed with 99% isopropanol containing 1% TFA and eluted with 80% acetonitrile containing 2% ammonium hydroxide followed by vacuum concentration. Peptides were resuspended in 2% acetonitrile containing 0.1% TFA and a 5% aliquot of each sample pooled and fractionated into 12 fractions as described above to generate a spectral library. The sample and MS file identifies have been uploaded as a table with the .raw proteomic data to the PRIDE ProteomeXchange (see Data availability section). Muscle tissue from rAAV6-treated mice were processed using the identical procedure described above with only minor modifications including the 10-plex TMT lot #WC306775 was used. The sample identity and labelling channels have been uploaded as a table with the .raw proteomic data to the PRIDE ProteomeXchange (see Data availability section). Peptides were fractionated using a separate in-house fabricated column with identical dimensions and particles, but a Dionex 3500 HPLC was used with the same detection at 210 nm and a total of 48 fractions collected and concatenated down to 12 fractions.

## Mass spectrometry and data processing

Peptide fractions from skeletal muscle of the HMDP were resuspended in 2% acetonitrile containing 0.1% TFA and analyzed on a Dionex ultra-high pressure liquid chromatography system coupled to an Orbitrap Lumos mass spectrometer. Briefly, peptides were separated on 40 cm × 75 µm column containing 1.9 um C18AQ Reprosil particles on a linear gradient of 2–30% acetonitrile over 2 hr. Electrospray ionization was performed at 2.3 kV with 40% RF lens and positively charged peptides detected via a full-scan MS (350–1550 m/z, 1e6 AGC, 60 K resolution, 50 ms injection time) followed by data-dependent MS/MS analysis performed with CID of 35% normalized collision energy (NCE) (rapid scan rate, 2e4 AGC, 50 ms injection time, 10 ms activation time, 0.7 m/z isolation) of the top 10 most abundant peptides. Synchronous-precursor selection with MS3 (SPS-MS3) analysis was enabled with HCD of 60 NCE (100–500 m/z, 50 K resolution, 1e5 AGC, 105 ms injection time) (*McAlister et al., 2014*). Dynamic exclusion was enabled for 60 s. Data were processed with Proteome Discoverer v2.3 and searched against the Mouse UniProt database (November 2018) using SEQUEST (*Eng et al., 1994*). The precursor MS tolerance was set to 20 ppm and the MS/MS tolerance was set to 0.8 Da with a maximum of two miss-cleavage. The peptides were searched with oxidation of methionine set as variable modification, and TMT on peptide N-terminus/lysine and carbamidomethylation of cysteine set as a fixed modification. All data was searched as a single batch and the peptide spectral matches (PSMs) of each database search filtered to 1% FDR using a target/decoy approach with Percolator (*Käll et al., 2007*). The filtered PSMs from each database search were grouped and q-values generated at the peptide level with the Qvality algorithm (*Käll et al., 2009*). Finally, the grouped peptide data was further filtered to 1% protein FDR using Protein Validator. Quantification was performed with the reporter ion quantification node for TMT quantification based on MS3 scans in Proteome Discoverer. TMT precision was set to 20 ppm and corrected for isotopic impurities. Only spectra with <50% co-isolation interference were used for quantification with an average signal-to-noise filter of >10. The data was filtered to retain Master proteins that were measured in at least 50 mice. Peptides from skeletal muscle micro-muscles were resuspended in 2% acetonitrile containing 0.1% TFA and analyzed on a Dionex ultra-high pressure liquid chromatography system coupled to an Orbitrap Exploris 480 mass spectrometer. Briefly, peptides were separated on 40 cm × 75 µm column containing 1.9 µm C18AQ

Reprosil particles on a linear gradient of 2–30% acetonitrile over 70 min. Electrospray ionization was performed at 1.9 kV with 40% RF lens and positively charged peptides detected via a full-scan MS (350–950 m/z, 2.5e6 AGC, 60 K resolution, 50 ms injection time) followed by data-independent MS/MS analysis performed with HCD of 28% NCE (16 m/z isolation, 38 windows with 1 m/z overlap, 2e6 AGC, 30 K resolution, auto injection time). The pooled and fractionated samples were used to generate a spectral library using data-dependent acquisition acquired in the same batch using the identical liquid chromatography and column. Each of the 12 fractions were injected twice using two-step gas-phase fraction to generate a spectral library. A full-scan MS from 350 to 651 m/z or 650 to 950 m/z was performed for each of the two injections (2.5e6 AGC, 60 K resolution, 50 ms injection time) followed by data-dependent MS/MS analysis performed with HCD of 28% NCE (1.2 m/z isolation, 5e4 AGC, 15 K resolution, auto injection time). Data were processed with Spectronaut v15.0.210615.50606 and the DDA data were searched against the Human UniProt database (June 2021) using Pulsar. The minimum peptide length set to seven amino acids with specific trypsin cleavage and search criteria included oxidation of methionine and protein N-terminal acetylation set as variable modifications, and carbamidomethylation set as a fixed modification. Data were filtered to 1% FDR at the peptide and protein level (q-value cut-off <0.01). The DIA data were searched within Spectronaut using the project-specific library and peptide quantification was performed at MS2 level using three to six fragment ions which included automated interference fragment ion removal as previously described (*Bruderer et al., 2015*). Dynamic mass MS1 and MS2 mass tolerance was enabled, and local (non-linear) regression was performed for retention time calibration. A dynamic extracted ion chromatogram window size was performed, and protein quantification performed with weighted peptide average values. Peptide fractions from skeletal muscle treated with rAAV6 were analyzed as described above for muscle of the HMDP with minor modifications. Briefly, peptides were separated using a Dionex ultra-high pressure liquid chromatography system using the identical chromatography configuration, but detection was achieved with an Orbitrap Eclipse mass spectrometer. Electrospray ionization was performed at 1.9 kV with 30% RF lens and positively charged peptides detected via a full-scan MS (350–1550 m/z, 2e6 AGC, 60 K resolution, 50 ms injection time) followed by data-dependent MS/MS analysis performed with HCD of 36% NCE (1e5 AGC, 86 ms injection time, 0.7 m/z isolation) with a 2.5 s cycle time and dynamic exclusion was enabled for 60 s. Data were processed with Proteome Discoverer v2.3 and searched against the Mouse UniProt database (February 2022) using SEQUEST (*Eng et al., 1994*). The precursor MS tolerance was set to 20 ppm and the MS/MS tolerance was set to 0.02 Da with a maximum of two miss-cleavage. The peptides were searched with oxidation of methionine set as variable modification, and TMT on peptide N-terminus/lysine and carbamidomethylation of cysteine set as a fixed modification. All data was searched as a single batch and the PSMs of each database search filtered to 1% FDR using a target/decoy approach with Percolator (*Käll et al., 2007*). The filtered PSMs from each database search were grouped and q-values generated at the peptide level with the Qvality algorithm (*Käll et al., 2009*). Finally, the grouped peptide data was further filtered to 1% protein FDR using Protein Validator. Quantification was performed with the reporter ion quantification node for TMT quantification based on MS2 scans in Proteome Discoverer. TMT precision was set to 20 ppm and corrected for isotopic impurities. Only spectra with <50% co-isolation interference were used for quantification with an average signal-to-noise filter of >10. The data was filtered to retain Master proteins that were measured in at least 50 mice.

## Protein-protein and protein-trait correlations

Data analyses were performed using R (version 4.1.1). Circos plots and circular dendrograms were created using the circlize R package (*Gu et al., 2014*). CoffeeProt was used to assess protein-protein correlations and to produce network plots (*Molendijk et al., 2021b*). TeaProt was used to perform functional enrichment analyses (*Molendijk et al., 2022*). The packages Gviz and TxDb.Mmusculus. UCSC.mm10.knownGene were used to produce genomic tracks (*Hahne and Ivanek, 2016*). Biweight midcorrelation (bicor) was performed using the WGCNA package (*Langfelder and Horvath, 2008*). Both the proteomics and trait datasets were summarized (mean) at the strain level prior to performing protein-trait correlations. The correlation coefficient and p-value were reported for each protein-trait pair, followed by the calculation of adjusted p-values (q-value) using the Benjamini-Hochberg procedure. Orthogonal partial least-squares (OPLS) modelling was performed using the ropls package (*Thévenot et al., 2015*) where the summarized proteomics data represents the input numerical matrix

(x), and the measurement of a trait of interest represents the response to be modelled (y). Models were created with a single predictive component (predI) and a single orthogonal component (orthoI). For OPLS and protein-trait correlations, proteomic data was quantile normalized and the biological replicates within each strain averaged. MuscleProt can be used to export the metrics, loadings, and scores tables from each model.

Protein and molecular/phenotypic quantitative trait locus (QTL) mapping The identification of SNPs associated to protein abundance was performed using an efficient mixed-model association (fast-lmm) (*Kang et al., 2008*) as described below where the model was adjusted for population structure (*Flint and Eskin, 2012*):

$$y = 1n\mu + x\beta + u + e \tag{1}$$

in which n is the number of individuals; μ is the mean; β is the allele effect of the SNP; x is the (n×1) vector of observed genotypes of the SNP. This model takes population structure into account, as u is the random effects due to genetic relatedness with var.(u)=σ2uK, and e denotes the random noise with var.(e)=σ2eI. Here, K indicates the identity-by-state kinship matrix estimated using all SNPs; I represents the (n×n) identity matrix; and 1n is the (n×1) vector of ones. σ2u and σ2e were estimated using restricted maximum likelihood and computed p-values using the standard F-test to test the null hypothesis in which β = 0. Genome-wide significance threshold and genome-wide association mapping were determined as the family-wise error rate as the probability of observing one or more false positives across all SNPs for a given phenotype. To correct for false discovery, q-values were estimated from the distribution of p-values using the linear mixed model from the R package 'q value'. Significance was calculated at q-value <0.1 (*cis*-pQTL = ±10 Mb of the gene, approximated local adjusted $p<1 \times 10^{-4}$, and *trans*-pQTL=approximated global adjusted $p<5 \times 10^{-8}$) as described previously (*Chick et al., 2016*; *Parker et al., 2019*). SNP locations and variant effects were retrieved from the Ensembl Variant database (release 102, GRCm38). Molecular/phenotypicQTL data were obtained from previously published studies and processed using fast-lmm as described above (*Ghazalpour et al., 2012*; *Ghazalpour et al., 2014*; *Parks et al., 2015*; *Norheim et al., 2017*; *Rau et al., 2017*; *Norheim et al., 2019*; *Parker et al., 2019*; *Tuominen et al., 2021*; *Norheim et al., 2021*). A summary of phenotypic data and sources is described in *Supplementary file 4*.

## Structural biology

The PROVEAN (Protein Variant Effect Analyzer) tool was used to predict the functional effects of missense mutations in our dataset (*Choi and Chan, 2015*). The Colabfold platform utilizing Alphafold to predict the protein structures was used generate the EPHX1 structure (*Jumper et al., 2021*; *Mirdita et al., 2022*). The mmseq2 method was used for the multiple sequence alignment step. Alphafold models were ranked by pLDDT and further assessed using the sequence coverage, sequence identity, and predicted alignment error metrics generated in Colabfold (*Mirdita et al., 2022*). Generated models were visualized using PyMol. The PROVEAN (*Choi et al., 2012*) and PolyPhen-2 (*Adzhubei et al., 2013*) servers were used to predict the functional effect of mutations. FoldX was used to determine the effects of mutations on protein stability (*Delgado et al., 2019*). DynaMut was used to determine the protein flexibility as a result of mutation and predict the interactomic interactions (*Rodrigues et al., 2018*). Mol* (Molstar) was used to display PDB format structures and colour protein complex constituents according to relative fold change values (*Sehnal et al., 2021*). Structural models for the SRP (7OBQ) and ribosome-SEC61 complex (3J7R) were retrieved from the RCSB Protein Data Bank (PDB). Models were edited to remove RNA and small molecule entities. The diagram of the TRAP complex was based on the electron microscopy density model EMD-3068. The R functions colorRampPalette and col2rgb were used to generate gradients of hex colour codes and corresponding RGB colour values used in Mol*.

## Database

SNP locations and variant effects were retrieved from the mus_musculus_incl_consequences.vcf.gz (03 August, 2020) file in the Ensembl Variant database (release 102) (*McLaren et al., 2016*). Gene information and UniProt accession mappings were retrieved from the Ensembl project, release 102 (*Howe et al., 2021*) Mus_musculus.GRCm38.102.gtf.gz (27 October 2020) and Mus_musculus.GRCm38.102.

uniprot.tsv.gz (26 October 2020). UK Biobank exome sequencing data was accessed through the Genebass webserver (*Karczewski et al., 2022*).

## shRNA:rAAV6 production

All shRNA sequences are shown in *Supplementary file 7* and were designed using SplashRNA including optimized stem-loop design (5'TAGTGAAGCCACAGATGTA) as previously described (*Pelossof et al., 2017*). The scramble shRNA sequence was 5'GATCGAATGTGTACTTCGA and selected based on a previously described screen for low toxicity (*Grimm et al., 2006*). All AAV vectors were produced by the Vector and Genome Engineering Facility (VGEF) at Children's Medical Research Institute (CMRI). Vectors were produced by standard transient transfection of 5×15 cm plates of HEK293 (ATCC# CRL-1573) cells using PEI (polyethylenimine, PolyPlus, Cat# 115-100) with a 1:1:2 molar ratio of pTransgene:pRep2CapXHelper:pAd5Helper. Vectors were purified using iodixanol gradient ultracentrifugation as previously described (*Khan et al., 2011*). Amicon Ultra-4 Centrifuge Filter Units (Ultracel-100 kDa membrane, EMD Millipore, Cat# UFC810024) were used to perform buffer exchange (phosphate-buffered saline [PBS, Gibco, Cat# 14190], 50 mM NaCl [Sigma-Aldrich, Cat# S5150-1L], 0.001%, Pluronic F68 [v/v] [Gibco, Cat# 24040]) and the final concentration step. Iodixanol-purified AAVs were quantified using droplet digital PCR (ddPCR [Bio-Rad, Berkeley]) using QX200 ddPCR EvaGreen Supermix (Cat# 1864034; Bio-Rad) with eGFP primers (5' TCAAGATCCGCC ACAACATC and 5' TTCTCGTTGGGGTCTTTGCT). All cell stocks were regularly checked for absence of mycoplasma with the Mycoplasma Detection Kit (Jena Bioscience; # PP-401).

## hµM production and functional assessment

hµMs were generated as described previously (*Mills et al., 2019*). Briefly, primary human skeletal muscle myoblasts from a male, 20 years of age (Lonza, lot #18TL269121) were mixed with collagen I gel to make a 3.5 µl final solution containing 3.3 mg/mL collagen I and 22% (v/v) Matrigel (52,500 cells per hµM). The bovine acid-solubilized collagen I (Devro) was first salt balanced and pH neutralized using 10× DMEM and 0.1 M NaOH, respectively, prior to mixing with Matrigel and then combined with the cells. The mixture was prepared on ice and pipetted into the cell-culture inserts (*Mills et al., 2017*). The mixture was then gelled at 37°C for 30 min. After 1 day of formation, media was switched to containing MEM α (Thermo Fisher Scientific) with 1% P/S (Thermo Fisher Scientific), and 1% B-27 supplement (Thermo Fisher Scientific), with 10 µM DAPT (Stem Cell Technologies) and 1 µM Dabrafenib (Stem Cell Technologies) to induce differentiation. On day 8, media was switched to maintenance media containing MEM α (Thermo Fisher Scientific) with 1% P/S (Thermo Fisher Scientific), and 1% B-27 supplement (Thermo Fisher Scientific). Media was changed every 2–3 days. On day 12, hµMs were treated with AAV6 encoding shRNA for genes of interest or a scrambled control at 6e7 vg/hµM. After 72 hr of treatment, hµM were analysed for their function via electrically stimulation at 10 or 20 Hz; 5 ms square pulses with 20 mA current using a Panlab/Harvard Apparatus Digital Stimulator. During stimulation, a Leica DMi8 inverted high content Imager was used to capture a 5 or 15 s time-lapse of each hµM contracting in real time at 37°C. Pole deflection was used to approximate the force of contraction as per *Mills et al., 2017*. Custom batch processing files were written in Matlab R2013a (Mathworks) to convert the stacked TIFF files to AVI, track the pole movement (using vision. PointTracker), produce a force-time figure, and export the batch data to an Excel (Microsoft) spreadsheet. hµM max force was assessed as the peak force of contraction during a 20 Hz stimulation for 1 s. Whilst, hµM endurance/fatigue was assessed as the change in contraction force in response to a 10 Hz stimulation for 10 s (force at the end of stimulation [10 s] compared to peak force).

## Mouse housing and rAAV6 intramuscular injection

All mouse experiments were approved by The University of Melbourne Animal Ethics Committee (AEC ID1914940) and conformed to the National Health and Medical Research Council of Australia guidelines regarding the care and use of experimental animals. C57BL/6J mice (JAX 000664) were obtained from Animal Resource Centre (WA, Australia). Mice were housed at 22°C (±1°C) in groups of five/cage and maintained on a Standard Chow diet (Specialty Feeds, Australia) with a 12 hr light/dark cycle and ad libitum access to food and water. For intramuscular injections of rAAV6, mice were anaesthetized with isoflurane (4% in oxygen at 1 l/min) and then transferred to a dissecting microscope stage with heat pad and isoflurane inhalation nose piece (2% in oxygen at 1 l/min). Unconsciousness

was assessed via the lack of leg and optical reflexes for at least 1 min to ensure head position does not affect normal breathing. Mice received subcutaneous analgesic injection of meloxicam between the shoulder blades (5 mg/kg) and the surface of the hindlimbs were sterilized with 80% ethanol. The TA/EDL muscles were injected with 2×1010 vector genomes/30 µl of rAAV6 using a 32 G needle. Mice were returned to cages and body weights monitored daily for the first 3 days and then weekly.

## Ex vivo muscle function testing

Mice were anaesthetized in the non-fasted state with isoflurane (4% in oxygen at 1 l/min; muscle contraction experiments) and transferred to a dissecting microscope stage with isoflurane inhalation nose piece (2% in oxygen at 1 l/min). Depth of anaesthesia was assessed via the lack of leg and optical reflexes for at least 1 min to ensure head position did not affect normal breathing. After confirming anaesthesia, skin from the hind legs was removed, and EDL muscles were sutured using 5.0 braided suture at both proximal and distal ends at the tendomuscular junction. Muscles were excised and incubated in Modified Krebs Buffer (116 mM NaCl, 4.6 mM KCl, 1.16 mM $KH_2PO_4$, 25.3 mM $NaHCO_3$, 2.5 mM $CaCl_2$, 1.16 mM $MgSO_4$) in a myograph (DMT, Denmark; #820 MS) at 30°C with constant gentle bubbling of 5% medical carbon dioxide in oxygen. For contractile function experiments, a DMT CS4 stimulator was used to deliver 0.2 ms supramaximal (26 V) pulses via stimulation electrodes (DMT 300145) placed over the mid-belly of the muscle. Successive twitch stimulations, with at least 30 s rest, were used to determine muscle optimal length by very carefully stretching the muscle to optimum length when maximal twitch force was obtained. The frequency-force relationship was determined by stimulating muscles at different frequencies (10–200 Hz, 350 ms duration) with muscles rested for 2 min between stimuli to avoid fatigue. Where appropriate, force values were normalized to muscle CSA (i.e. to calculate specific force) by diving muscle mass by the product of muscle length and muscle density (1.06 mg/mm$^3$). A PowerLab 8/35 unit (ADInstruments) was used to digitize all force recordings and the Peak Parameters module in LabChart Pro (v8.1.16, ADInstruments) used for analysis of force responses.

## ALS model mice

SOD1G37R mice were sourced from The Jackson Laboratory and were subsequently bred and housed at the University of Melbourne, Melbourne, as previously described (*Roberts et al., 2014*) to generate transgenic mice and non-transgenic littermates used in this study. Animals were sacrificed at 25 weeks of age and tissues collected using previously described protocols in the non-fasted state (*Hilton et al., 2017*). All studies involving the use of SOD1G37R mice and non-transgenic littermates were approved by a University of Melbourne Animal Experimentation Ethics Committee (approval #2015124) and conformed with guidelines of the Australian National Health and Medical Research Council.

## Western blotting

Proteins were separated on NuPAGE 4–12% Bis-Tris protein gels (Thermo Fisher Scientific) in MOPS SDS Running Buffer at 160 V for 1 hr at room temperature. The protein was transferred to PVDF membranes (Millipore; #IPFL00010) in NuPAGE Transfer Buffer at 20 V for 1 hr at room temperature and blocked with 5% skim milk in Tris-buffered saline containing 0.1% Tween-20 (TBST) for at least 30 min at room temperature with gentle shaking. The membranes were incubated overnight in primary antibody with 5% BSA in TBST with gentle shaking at 4°C and washed three times in TBST at room temperature. Anti-UFC1 (EPR15014-102, ab189252), anti-UFSP2 (EP13424-49, ab192597), and anti-UFM1 (EPR4264(2), ab109305) were ordered from Abcam. The membranes were incubated with HRP-secondary antibody in 5% skim milk in TBST for 45 min at room temperature and washed three times with TBST. HRP-Donkey Anti-Rabbit (711-035-152, RRID:AB_10015282) was ordered from Jackson ImmunoResearch. Protein was visualized with Immobilon Western Chemiluminescent HRP Substrate (Millipore; #WBKLS0500) and imaged on a ChemiDoc (Bio-Rad). Densitometry was performed in ImageJ (*Schneider et al., 2012*).

## Immunostaining and microscopy

TA muscles were embedded in optimal cutting temperature compound (Tissue-Tek) and frozen in 2-methylbutane (Sigma-Aldrich; #320404) cooled in liquid nitrogen. Eight µm serial transverse sections were cut from the middle of the TA in a cryostat then mounted on uncoated, pre-cleaned glass slides.

Sections were fixed in 4% PFA for 10 min then blocked for 1 hr at room temperature in goat serum solution (5% goat serum, 2% BSA, 0.1% Triton in PBS). Sections were then air-dried and incubated in a humidity chamber overnight at room temperature in a primary antibody cocktail solution comprised of 1:25 SC-71 (DSHB; mouse IgG1), 1:10 BF-F3 (DSHB; mouse IgM), and 1:250 Laminin (Sigma; L9393; rabbit IgG) in 0.05% PBSTween-20 to differentiate MHC type I, MHC type IIA, MHC type IIB fibers, and laminin regions, respectively. All non-reactive fibers were assumed to be MHC type IIX fibers. After primary incubation, sections were washed three times with PBS then incubated in a humidity chamber for 1.5 hr at room temperature in a secondary antibody cocktail solution comprised of 1:250 Alexa Fluor 555 (Goat anti-mouse IgG1), 1:250 Alexa Fluor 350 (Goat anti-mouse IgM), and 1:250 Alexa Fluor 647 (Goat anti-rabbit IgG) in 0.05% PBSTween-20. After secondary incubation, sections were washed three times with PBS, air-dried and mounted with Fluoro-Gel (ProSciTech IM030) under a coverslip. Fluorescence imaging of the whole section was captured with an upright microscope with a camera (Axio Imager M2, Carl Zeiss, Wrek, Göttingen, Germany). Pseudo colouring of each fluorescence channel was performed with ZEN 3.3 (Blue edition, Carl Zeiss, Wrek, Göttingen, Germany). Quantification was performed with Fiji (ImageJ; NIH).

## MuscleProt web server implementation

MuscleProt was developed using the R programming language for the backend and relies on the shiny package for the web server front-end in addition to HTML, CSS, and JavaScript. The WGCNA package is used to perform biweight midcorrelation (bicor) analyses (*Langfelder and Horvath, 2008*) and interactive network plots are created using networkD3. Multivariate analyses are performed using the ropls package (*Thévenot et al., 2015*). Tables are created using the DT package and all other visualizations use a combination of ggplot2 and plotly. MuscleProt is deployed on the Melbourne Research Cloud, running Ubuntu 18.04 and utilizing hypervisors built on AMD EPYC 2 (base CPU clock speed 2.0 GHz, burst clock speed 3.35 GHz). The Melbourne Research Cloud is based on the OpenStack open-source cloud platform.

## Statistical analysis

Statistics on densitometry, muscle function, and histology were performed in GraphPad Prism (Version 9.0.0). T-tests (unpaired for ALS analysis or paired for rAAV6 analysis) or two-way ANOVA (rAAV6 analysis) were used with a significance level of $p < 0.05$. All mentions of sample size (n) refer to biological replicates.

## Acknowledgements

We would like to thank the Melbourne Mass Spectrometry and Proteomics Facility in The Bio21 Molecular Science and Biotechnology Institute at The University of Melbourne, and the Sydney Mass Spectrometry Facility in the Charles Perkins Centre at The University of Sydney for mass spectrometry support. We also thank the Melbourne Mouse Metabolic Phenotyping Platform at the University of Melbourne, and facilities at the University of California, Los Angles for mouse support. This research was supported by use of the Nectar Research Cloud and by the University of Melbourne Research Platform Services. The Nectar Research Cloud is a collaborative Australian research platform supported by the National Collaborative Research Infrastructure Strategy.

## Additional information

### Competing interests
David E James: Senior editor, *eLife*. The other authors declare that no competing interests exist.

## Funding

| Funder | Grant reference number | Author |
|---|---|---|
| National Health and Medical Research Council | APP1184363 | Karen Reue<br>Marcus M Seldin<br>Benjamin L Parker |
| National Health and Medical Research Council | APP2009642 | Benjamin L Parker |
| National Health and Medical Research Council | APP2013189 | Richard J Mills |
| National Health and Medical Research Council | APP1156562 | Paul Gregorevic<br>Benjamin L Parker |
| National Institutes of Health | HL138193 | Marcus M Seldin |
| National Institutes of Health | DK130640 | Marcus M Seldin |
| National Institutes of Health | DK097771 | Marcus M Seldin |
| National Institutes of Health | GM115318 | Karen Reue |
| National Institutes of Health | AG070959 | Aldons J Lusis |
| National Institutes of Health | HL147883 | Aldons J Lusis |
| National Institutes of Health | DK117850 | Aldons J Lusis |
| Weary Dunlop Foundation | | Benjamin L Parker |
| The ALS Association | 21-DDC-574 | Paul Gregorevic<br>Peter J Crouch |

The funders had no role in study design, data collection and interpretation, or the decision to submit the work for publication.

## Author contributions

Jeffrey Molendijk, Software, Formal analysis, Visualization, Writing – original draft, Writing – review and editing; Ronnie Blazev, Formal analysis, Investigation, Methodology, Writing – review and editing; Richard J Mills, Formal analysis, Investigation, Writing – review and editing; Yaan-Kit Ng, Formal analysis, Writing – review and editing; Kevin I Watt, Peter J Crouch, Investigation, Writing – review and editing; Daryn Chau, Paul Gregorevic, James BW Hilton, Leszek Lisowski, Peixiang Zhang, Karen Reue, James E Hudson, Methodology, Writing – review and editing; Aldons J Lusis, David E James, Conceptualization, Writing – review and editing; Marcus M Seldin, Conceptualization, Formal analysis, Investigation, Methodology, Writing – review and editing; Benjamin L Parker, Conceptualization, Formal analysis, Investigation, Visualization, Methodology, Writing – original draft, Writing – review and editing

## Author ORCIDs

Jeffrey Molendijk ![ORCID] http://orcid.org/0000-0001-6575-504X
Peter J Crouch ![ORCID] http://orcid.org/0000-0002-7777-4747
David E James ![ORCID] http://orcid.org/0000-0001-5946-5257
Marcus M Seldin ![ORCID] http://orcid.org/0000-0001-8026-4759
Benjamin L Parker ![ORCID] http://orcid.org/0000-0003-1818-2183

### Ethics

All rAAV6 intramuscular injection mouse experiments were approved by The University of Melbourne Animal Ethics Committee (AEC ID1914940) and conformed to the National Health and Medical Research Council of Australia guidelines regarding the care and use of experimental animals. All studies involving the use of SOD1G37R mice and non-transgenic littermates were approved by a University of Melbourne Animal Experimentation Ethics Committee (approval #2015124) and conformed with guidelines of the Australian National Health and Medical Research Council.

### Decision letter and Author response

Decision letter https://doi.org/10.7554/eLife.82951.sa1
Author response https://doi.org/10.7554/eLife.82951.sa2

## Additional files

### Supplementary files

• Supplementary file 1. Hybrid Mouse Diversity Panel (HMDP) skeletal muscle proteomics. Proteomics data of gastrocnemius muscle displaying quantification ratios of each sample compared to its corresponding pooled tandem mass tag (TMT) control. PEP: posterior error probability. Related to *Figures 1–3*, *Figure 1—figure supplement 1*.

• Supplementary file 2. Hybrid Mouse Diversity Panel (HMDP) skeletal muscle protein-quantitative trait loci (pQTLs). Protein quantitative trait loci from 161 HMDP cohort mice. Table contains *cis*-pQTLs ($p < 1 \times 10^{-4}$) and *trans*-pQTLs ($p < 5 \times 10^{-8}$), including genomic locations of the single nucleotide polymorphism (SNP) and associated gene. pQTLs are annotated with proxy (*cis/trans*), intragenic variants, known LD blocks, variant effect, variant impact, and target screen prioritization columns. Related to *Figures 1–3*, and *Figure 1—figure supplement 2*.

• Supplementary file 3. Hybrid Mouse Diversity Panel (HMDP) skeletal muscle pairwise protein-protein correlations. Protein-protein correlation as determined using biweight midcorrelation. p-Values and q-values derived using the Benjamini-Hochberg procedure, and only positive correlations are shown (cor > 0.3 and q < 0.05). Correlated protein pairs are annotated with protein:protein interactions from the CORUM and BioPlex databases, and subcellar. Related to *Figures 1 and 2*.

• Supplementary file 4. Hybrid Mouse Diversity Panel (HMDP) molecular or phenotypic traits. Summary of traits integrated into the current study including Pubmed ID sources. Related to *Figures 2 and 3*.

• Supplementary file 5. Hybrid Mouse Diversity Panel (HMDP) molecular or phenotypic quantitative trait loci (QTLs). Table contains QTLs ($p < 1 \times 10^{-4}$) including chromosome, genomic location, including Pubmed ID sources. Related to *Figures 2 and 3*.

• Supplementary file 6. Proteomics of skeletal muscle treated with either rAAV6:shScramble or AAV6:shUFC1. Proteomics of extensor digitorum long (EDL) muscles displaying tandem mass tag (TMT) quantification expressed as Log2(area under the curve). Significance was calculated using paired Student's t-test with Benjamini-Hochberg FDR. PEP: posterior error probability. Related to *Figure 4*.

• Supplementary file 7. shRNA sequences used in the human micro-muscle screen and mouse shUFC1 experiments. Related to *Figure 4*.

• MDAR checklist

• Source data 1. Genebass datasets for UFC1, EPHX1, DUSP23, NIT1, MPZ, BPNT1 and PCP4L1.

### Data availability

The proteomics data generated in this study are deposited to the ProteomeXchange Consortium via the PRIDE (*Perez-Riverol et al., 2019*) under the identifiers PXD032729, PXD034913 and PXD035170. The code used for downstream analysis of proteomic data can be found at: https://github.com/JeffreyMolendijk/skeletal_muscle, (copy archived at swh:1:rev:9311d7b-fb59979d80e18612879631dc78f2f0902; *Molendijk, 2022*). The following entries

from Genebass (*Karczewski et al., 2022*) were used: UFC1, EPHX1, DUSP23, NIT1, MPZ, BPNT1 and PCP4L1.

The following datasets were generated:

| Author(s) | Year | Dataset title | Dataset URL | Database and Identifier |
|---|---|---|---|---|
| Parker BL | 2023 | Proteomic analysis of a targeted functional genomic screen in human skeletal muscle organoids | https://www.ebi.ac.uk/pride/archive/projects/PXD034913 | PRIDE, PXD034913 |
| Parker BL | 2023 | Proteomic analysis of UFC1 knockdown in mouse skeletal muscle | https://www.ebi.ac.uk/pride/archive/projects/PXD035170 | PRIDE, PXD035170 |
| Parker BL | 2023 | Proteomic analysis of skeletal muscle from the Hybrid Mouse Diversity Panel | https://www.ebi.ac.uk/pride/archive/projects/PXD032729 | PRIDE, PXD032729 |

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

# Appendix 1

### Appendix 1—key resources table

| Reagent type (species) or resource | Designation | Source or reference | Identifiers | Additional information |
|---|---|---|---|---|
| Antibody | Rabbit monoclonal anti-UFC1 | Abcam | EPR15014-102 (ab189252) | (1:1000) |
| Antibody | Rabbit monoclonal anti-UFSP2 | Abcam | EP13424-49 (ab192597) | (1:1000) |
| Antibody | Rabbit monoclonal anti-UFM1 | Abcam | EPR4264(2) (ab109305) | (1:1000) |
| Antibody | Rabbit monoclonal antiBiP | Cell Signaling Technologies | 3177 | (1:1000) |
| Antibody | Rabbit monoclonal SQSTM1/p62 (D1Q5S) | Cell Signaling Technologies | 39749 | (1:1000) |
| Antibody | Rabbit monoclonal K48-linkage Specific Polyubiquitin (D9D5) | Cell Signaling Technologies | 8081 | (1:1000) |
| Antibody | Donkey polyclonal Anti-Rabbit-HRP | Jackson ImmunoResearch | 711-035-152 (RRID:AB_10015282) | (1:10000) |
| Genetic reagent (*Homo sapiens*) | Human Skeletal Myoblasts | Lonza | CC-2580 (lot #18TL269121) | |
| Genetic reagent (*Homo sapiens*) | Human embryonic kidney 293 cells expressing SV40 large T antigen | ATCC | CRL-1573 | |
| Strain, strain background (*Mus musculus*) | A/J | JAX | RRID:IMSR_JAX:000646 | |
| Strain, strain background (*Mus musculus*) | AXB10/PgnJ | JAX | RRID:IMSR_JAX:001681 | |
| Strain, strain background (*Mus musculus*) | AXB13/PgnJ | JAX | RRID:IMSR_JAX:001684 | |
| Strain, strain background (*Mus musculus*) | AXB15/PgnJ | JAX | RRID:IMSR_JAX:001685 | |
| Strain, strain background (*Mus musculus*) | AXB19a/PgnJ | JAX | RRID:IMSR_JAX:001686 | |
| Strain, strain background (*Mus musculus*) | AXB4/PgnJ | JAX | RRID:IMSR_JAX:001676 | |
| Strain, strain background (*Mus musculus*) | AXB8/PgnJ | JAX | RRID:IMSR_JAX:001679 | |
| Strain, strain background (*Mus musculus*) | B6.Cg-Tg(SOD1*G37R)42Dpr/J | JAX | RRID:IMSR_JAX:008342 | |
| Strain, strain background (*Mus musculus*) | BALB/cByJ | JAX | RRID:IMSR_JAX:001026 | |
| Strain, strain background (*Mus musculus*) | BTBR T+tf/J | NA | NA | |
| Strain, strain background (*Mus musculus*) | BUB/BnJ | JAX | RRID:IMSR_JAX:000653 | |

*Appendix 1 Continued on next page*

*Appendix 1 Continued*

| Reagent type (species) or resource | Designation | Source or reference | Identifiers | Additional information |
|---|---|---|---|---|
| Strain, strain background (*Mus musculus*) | BXA12/PgnJ | JAX | RRID:IMSR_JAX:001700 | |
| Strain, strain background (*Mus musculus*) | BXA13/PgnJ | JAX | RRID:IMSR_JAX:001826 | |
| Strain, strain background (*Mus musculus*) | BXA14/PgnJ | JAX | RRID:IMSR_JAX:001702 | |
| Strain, strain background (*Mus musculus*) | BXA16/PgnJ | JAX | RRID:IMSR_JAX:001703 | |
| Strain, strain background (*Mus musculus*) | BXA2/PgnJ | JAX | RRID:IMSR_JAX:001693 | |
| Strain, strain background (*Mus musculus*) | BXA4/PgnJ | JAX | RRID:IMSR_JAX:001694 | |
| Strain, strain background (*Mus musculus*) | BXD100 | NA | NA | |
| Strain, strain background (*Mus musculus*) | BXD100/RwwJ | JAX | RRID:IMSR_JAX:007143 | |
| Strain, strain background (*Mus musculus*) | BXD12/TyJ | JAX | RRID:IMSR_JAX:000045 | |
| Strain, strain background (*Mus musculus*) | BXD14/TyJ | JAX | RRID:IMSR_JAX:000329 | |
| Strain, strain background (*Mus musculus*) | BXD19/TyJ | JAX | RRID:IMSR_JAX:000010 | |
| Strain, strain background (*Mus musculus*) | BXD21/TyJ | JAX | RRID:IMSR_JAX:000077 | |
| Strain, strain background (*Mus musculus*) | BXD22/TyJ | JAX | RRID:IMSR_JAX:000043 | |
| Strain, strain background (*Mus musculus*) | BXD27/TyJ | JAX | RRID:IMSR_JAX:000041 | |
| Strain, strain background (*Mus musculus*) | BXD28/TyJ | JAX | RRID:IMSR_JAX:000047 | |
| Strain, strain background (*Mus musculus*) | BXD29/TyJ | NA | NA | |
| Strain, strain background (*Mus musculus*) | BXD31/TyJ | JAX | RRID:IMSR_JAX:000083 | |
| Strain, strain background (*Mus musculus*) | BXD32/TyJ | JAX | RRID:IMSR_JAX:000078 | |

*Appendix 1 Continued on next page*

*Appendix 1 Continued*

| Reagent type (species) or resource | Designation | Source or reference | Identifiers | Additional information |
|---|---|---|---|---|
| Strain, strain background (*Mus musculus*) | BXD33/TyJ | JAX | RRID:IMSR_JAX:003222 | |
| Strain, strain background (*Mus musculus*) | BXD34/TyJ | JAX | RRID:IMSR_JAX:003223 | |
| Strain, strain background (*Mus musculus*) | BXD39/TyJ | JAX | RRID:IMSR_JAX:003228 | |
| Strain, strain background (*Mus musculus*) | BXD40/TyJ | JAX | RRID:IMSR_JAX:003229 | |
| Strain, strain background (*Mus musculus*) | BXD44/RwwJ | JAX | RRID:IMSR_JAX:007094 | |
| Strain, strain background (*Mus musculus*) | BXD45/RwwJ | JAX | RRID:IMSR_JAX:007096 | |
| Strain, strain background (*Mus musculus*) | BXD48/RwwJ | JAX | RRID:IMSR_JAX:007097 | |
| Strain, strain background (*Mus musculus*) | BXD48A | NA | NA | |
| Strain, strain background (*Mus musculus*) | BXD5/TyJ | JAX | RRID:IMSR_JAX:000037 | |
| Strain, strain background (*Mus musculus*) | BXD50/RwwJ | JAX | RRID:IMSR_JAX:007099 | |
| Strain, strain background (*Mus musculus*) | BXD51/RwwJ | JAX | RRID:IMSR_JAX:007100 | |
| Strain, strain background (*Mus musculus*) | BXD55/RwwJ | JAX | RRID:IMSR_JAX:007103 | |
| Strain, strain background (*Mus musculus*) | BXD60/RwwJ | JAX | RRID:IMSR_JAX:007105 | |
| Strain, strain background (*Mus musculus*) | BXD61/RwwJ | JAX | RRID:IMSR_JAX:007106 | |
| Strain, strain background (*Mus musculus*) | BXD62/RwwJ | JAX | RRID:IMSR_JAX:007107 | |
| Strain, strain background (*Mus musculus*) | BXD63 | NA | NA | |
| Strain, strain background (*Mus musculus*) | BXD65 | NA | NA | |
| Strain, strain background (*Mus musculus*) | BXD66/RwwJ | JAX | RRID:IMSR_JAX:007111 | |

*Appendix 1 Continued on next page*

*Appendix 1 Continued*

| Reagent type (species) or resource | Designation | Source or reference | Identifiers | Additional information |
|---|---|---|---|---|
| Strain, strain background (*Mus musculus*) | BXD67/RwwJ | JAX | RRID:IMSR_JAX:007112 | |
| Strain, strain background (*Mus musculus*) | BXD68/RwwJ | JAX | RRID:IMSR_JAX:007113 | |
| Strain, strain background (*Mus musculus*) | BXD69/RwwJ | JAX | RRID:IMSR_JAX:007114 | |
| Strain, strain background (*Mus musculus*) | BXD73/RwwJ | JAX | RRID:IMSR_JAX:007117 | |
| Strain, strain background (*Mus musculus*) | BXD75/RwwJ | JAX | RRID:IMSR_JAX:007119 | |
| Strain, strain background (*Mus musculus*) | BXD86/RwwJ | JAX | RRID:IMSR_JAX:007129 | |
| Strain, strain background (*Mus musculus*) | BXD87/RwwJ | JAX | RRID:IMSR_JAX:007130 | |
| Strain, strain background (*Mus musculus*) | BXH10/TyJ | JAX | RRID:IMSR_JAX:000032 | |
| Strain, strain background (*Mus musculus*) | BXH14/TyJ | JAX | RRID:IMSR_JAX:000009 | |
| Strain, strain background (*Mus musculus*) | BXH8/TyJ | JAX | RRID:IMSR_JAX:000076 | |
| Strain, strain background (*Mus musculus*) | C3H/HeJ | JAX | RRID:IMSR_JAX:000659 | |
| Strain, strain background (*Mus musculus*) | C57BL/6J | JAX | RRID:IMSR_JAX:000664 | |
| Strain, strain background (*Mus musculus*) | C58/J | JAX | RRID:IMSR_JAX:000669 | |
| Strain, strain background (*Mus musculus*) | CBA/J | JAX | RRID:IMSR_JAX:000656 | |
| Strain, strain background (*Mus musculus*) | CE/J | JAX | RRID:IMSR_JAX:000657 | |
| Strain, strain background (*Mus musculus*) | CXB12/HiAJ | JAX | RRID:IMSR_JAX:001633 | |
| Strain, strain background (*Mus musculus*) | CXB2/ByJ | JAX | RRID:IMSR_JAX:000352 | |
| Strain, strain background (*Mus musculus*) | DBA/2J | JAX | RRID:IMSR_JAX:000671 | |

*Appendix 1 Continued*

| Reagent type (species) or resource | Designation | Source or reference | Identifiers | Additional information |
|---|---|---|---|---|
| Strain, strain background (*Mus musculus*) | FVB/NJ | JAX | RRID:IMSR_JAX:001800 | |
| Strain, strain background (*Mus musculus*) | LG/J | JAX | RRID:IMSR_JAX:000675 | |
| Strain, strain background (*Mus musculus*) | LP/J | JAX | RRID:IMSR_JAX:000676 | |
| Strain, strain background (*Mus musculus*) | MRL/MpJ | JAX | RRID:IMSR_JAX:000486 | |
| Strain, strain background (*Mus musculus*) | NON/ShiLtJ | JAX | RRID:IMSR_JAX:002423 | |
| Strain, strain background (*Mus musculus*) | NOR/LtJ | JAX | RRID:IMSR_JAX:002050 | |
| Strain, strain background (*Mus musculus*) | NZB/BINJ | JAX | RRID:IMSR_JAX:000684 | |
| Strain, strain background (*Mus musculus*) | PL/J | JAX | RRID:IMSR_JAX:000680 | |
| Strain, strain background (*Mus musculus*) | SJL/J | JAX | RRID:IMSR_JAX:000686 | |
| Software, algorithm | R version 4.1.1 | R Development Core Team, 2016 | https://www.R-project.org/ | |
| Software, algorithm | Limma 3.32.2 | *Ritchie et al., 2015* | https://bioconductor.org/packages/release/bioc/html/limma.html | |
| Software, algorithm | CoffeeProt | *Molendijk and Parker, 2021a* | https://www.coffeeprot.com | |
| Software, algorithm | TeaProt | *Molendijk et al., 2022* | https://tea.coffeeprot.com | |
| Software, algorithm | Mol* (Molstar) | *Sehnal et al., 2021* | https://molstar.org/ | |
| Software, algorithm | WGCNA | *Langfelder and Horvath, 2008* | https://cran.r-project.org/web/packages/WGCNA/ | |
| Software, algorithm | ColabFold (Alphafold2) | *Mirdita et al., 2022* | https://colab.research.google.com/github/sokrypton/ColabFold/blob/main/AlphaFold2.ipynb | |
| Software, algorithm | Genebass | *Karczewski et al., 2022* | https://app.genebass.org/ | |
| Software, algorithm | FoldX | *Delgado et al., 2019* | http://foldxsuite.crg.eu/ | |
| Software, algorithm | PROVEAN | *Choi and Chan, 2015* | http://provean.jcvi.org/index.php | |
| Software, algorithm | DynaMut | *Rodrigues et al., 2018* | http://biosig.unimelb.edu.au/dynamut/ | |

