## [Editor Report]

This manuscript will be of broad interest to those working in the genetics of complex diseases, with the results strongly supporting the author's primary claims. Overall, this is an important study that demonstrates the power of proteomics-based systems genetics studies in the mouse.

---

## [Decision Letter]

**Decision letter after peer review:**

Thank you for submitting your article "Proteome-wide systems genetics identifies UFMylation as a regulator of skeletal muscle function" for consideration by *eLife*. Your article has been reviewed by 2 peer reviewers, and the evaluation has been overseen by a Reviewing Editor and Christian Landry as the Senior Editor. The reviewers have opted to remain anonymous.

The reviewers have discussed their reviews with one another, and the Reviewing Editor has drafted this to help you prepare a revised submission. We are delighted that the concerns of the reviewers are minor.

*Reviewer #2 (Recommendations for the authors):*

The authors have performed a large forward systems genetics approach of around 160 female mice from the HMDP family, then identified candidate genes related to muscle phenotypes. They have selected a couple dozen of these candidates for mechanistic validation, of which one – UFC1 – clearly worked at both the cell line level and in vivo in a mouse model. This study opens up a wide number of new questions about how this mechanism works, what other phenotypes it may affect, how generally may it apply across male mice or to humans, and so forth.

Comments, in approximate order of appearance

1) I think this is the first time I've ever seen a submitted paper for review in a nice, clean format, rather than a double-spaced raw Word file. I appreciate this and hope that this becomes the norm, replacing the format that we've inherited from the time of typewriters. Anyway, moving on.

2) Lines 64-66: "However, all these initial proteome-wide systems genetics studies in mouse GRPs have focused on liver…" That's true for the initial studies, but there are at least a couple of older cross-tissue proteomics papers that weren't on the liver, e.g. PMID: 29945935. Coincidentally that paper also measures skeletal muscle and quantified quite substantially fewer proteins in muscle than the other 4 tissues assayed, which fits with the relatively low number of proteins quantified in this study compared to what you now typically see with TMT proteomics. A similar finding was made in a more recent and still-unpublished (and just on B6 mice) study from the Churchill group, biorxiv DOI: 10.1101/2022.05.17.492125.

3) Line 114: the RPL7 cis-pQTL that also almost certainly explains RPL19 and RPL23 trans-pQTLs is nice to see, especially as I see there are no other ribosomal genes in the area. I know this is a protein-focused paper, but (a) do you see the RPL7 cis-pQTL in other tissues you have already analyzed, like liver, and (b) are there cis/trans-eQTLs for any of these specific genes in the HMDP for other datasets you have processed? While by itself this seems rather irrelevant to your paper, what I mean is for the phenotype/cis-pQTL pairs you do in Figure 3, i.e. connecting muscle cis-pQTLs to HOMA indices, it would be nice to have a ballpark estimate of how many cis-pQTLs you would even expect to see across-tissue at the cis-pQTL level. Somewhat less relevantly, it would also be nice to know how many of those are also cis-eQTL or trans-eQTL pairs, although I think that has been fairly extensively studied by now (off the top of my head trans-pQTL-trans-eQTL pair: very, very unlikely; cis-eQTL-cis-pQTL pair: maybe 30-50% probability).

4) Line 123: "Among the 527 proteins with a cis-pQTL association, 212 had an intragenic association." How was this determined? It would be nice to have a supplemental table list of all of these proteins and their variant.

5) Lines ~190-215: There are a lot of associations here between genes with cis-pQTLs and phenotypes which I wouldn't really associate so strongly with the muscle proteome in the first place like visceral fat. Of course, if a gene has a cis-pQTL in one tissue it's highly likely (~50% if I remember right) to have a cis-pQTL in any other tissue where it's also expressed. The authors do mention this caveat, e.g. on line 197 mentioned that the pancreas would make more sense to check for insulin concentration phenotypes. There's also the concern that in Figure 3B that we're looking at linear correlations here, and correlating any protein with something like plasma insulin or HOMA-IR can dramatically change depending on a million different variables. For instance, were these plasma insulin calculated in exactly the same individuals as in this muscle proteome study or just the same strains? If just the same strains are the fasted states performed in exactly the same way? I know everyone does this – and me too – I just wonder how reasonable it is. The other examples shown here that you actually follow up on later in the paper, like grip strength and lean mass make sense. One question here though: grip strength also tends to be a rather unreliable measurement if the data for the HMDP strains were not all performed by exactly the same technician and in the exact same mice as measured here. Are all these phenotypes from the same exact females from this study? Or just the same strains? On that note, what's the observed heritability for all of these traits? That might be nice and easy to have on coffeeprot. Of course, this whole figure is "just" hypothesis selection so the selection criteria don't need to be bulletproof (even if it is certainly better if they are), but for instance, I couldn't figure out if the phenotypes were from exactly the same mice as the proteome measurements.

(6) Which HMDP strains have the SNPs that cause the QTL in UFC1? Is it relatively common in the HMDP or only a few strains? If relatively common, e.g. if it segregates between B6 and D2 or between any of the CC founders, is the same effect seen more widely? If it is common in any strain used for mouse population studies, it would be feasible to indicate whether the mechanism here found in females is more general.

7) Figure 5 in general: This looks convincing. I would also imagine, even though it's not my field, that there are now many proteomics studies of muscle proteome expression in ALS mice, as well as mice with other skeletomuscular disorders (DMD?). It would be interesting to know if UFC1 – and the other targets as well – are differentially regulated in such cohorts, especially as the phenotypic effect size here with the shRNA is huge. What about humans with ALS? Checking muscle proteomes in humans and observing differences in UFC1 (or really any of these gene targets) would be a huge finding, and it should not be so hard to check this through collaborators with access to human muscle datasets for those patients with ALS, DMD, or what have you.

8) In a related question, so you have run proteomics on the EDL muscles of a few mice; do the other proteins identified in Figure 3 also change, i.e. are any of them among the top 573? The findings highlighted here are also particularly interesting to me as I doubt any of them would be visible at the mRNA level as they all are proteins involved in large complexes (SEC61, RPL, TRAP, …), which could well indicate why this mechanism has been missed before in literature despite it being a highly-studied phenotype. Conversely, since the theorized mechanism is for hitting histone H4 and a bunch of other machinery involved in both transcription and translation, I would expect to see some signal at the mRNA level, even if it's not the same set of genes at all as at the protein level.

9) I don't see the fasted state mentioned in any protocol except for the HMDP. For instance, the protocol for the ALS animals has the Hilton 2017 paper cited for "as previously described" but that paper doesn't appear to mention fasting. Same for the AAV study, and since this was done in Australia and the HMDP in UCLA, I imagine there are some differences, if not in fasting then perhaps in other criteria. This is only a tidbit here since the actual validation of the UFC1 target clearly worked, but it could be a much larger issue if the authors had, say, followed up on the plasma insulin phenotype highlighted in Figure 3.

10) This is not a protocol paper so I am not judging it off of coffeeprot, but I have some issues with the website. For instance, I wanted to download the grip strength data and calculate heritability myself, but when I download the data, it just has an average value per strain, no mention of n and SEM, nor all values per strain – just a simple vector. For correlations and mapping, it is very easy to use. I think I already asked this – but are the phenotypes listed here from exactly the same HMDP mice used for this study? Certainly, the utility of the HMDP is that they do not need to be, but if not I am wondering then for traits like "Liver Weight_fast" if this is males, females, mixed, …

11) I am a bit confused about Supplemental Table 1. It is great this is available, but it doesn't seem to match exactly the protein table I get from Coffeeprot. Also with the table downloaded from coffeeprot, I see it goes from -3.866 to +2.695 across 4027 proteins and 33 strains, but in Table S1, I get values from -6.78 to +6.35 across 5350 proteins from 161 cohorts. OK, I see that it appears to just be removing the strain and protein line if the phenotype was not measured in that strain, and removing proteins if they are NA in all of the strains that do have the phenotype measured. The confusion comes when I look at something that is paired; for instance, the expression of E9Q1W3. In Table S1, I get values of 0.13 and 0.07 for AXB10 (#51 and #52) but in the CoffeeProt file, I get just a single value of 0.154 for AXB10, which is not clearly related to the 0.13 and 0.07. There is probably an easy explanation here, but I'm not getting it. On a related note, the units for the figures are a bit confusing, e.g. Figure 1F has no units for the proteins, but FigureI1 has the units as log2AUC. Aren't they all ratios based on the TMT control? Or have additional transformations been carried out?

---

## [Author Response]

Reviewer #2 (Recommendations for the authors):The authors have performed a large forward systems genetics approach of around 160 female mice from the HMDP family, then identified candidate genes related to muscle phenotypes. They have selected a couple dozen of these candidates for mechanistic validation, of which one – UFC1 – clearly worked at both the cell line level and in vivo in a mouse model. This study opens up a wide number of new questions about how this mechanism works, what other phenotypes it may affect, how generally may it apply across male mice or to humans, and so forth.Comments, in approximate order of appearance1) I think this is the first time I've ever seen a submitted paper for review in a nice, clean format, rather than a double-spaced raw Word file. I appreciate this and hope that this becomes the norm, replacing the format that we've inherited from the time of typewriters. Anyway, moving on.

We thank the reviewer for bringing this to our attention. We have updated this section to include the additional reference mentioned by the reviewer, and also included an additional publication since first submission. Note, the final reference suggested by the reviewer did not perform analysis on a genetic reference panel (GRP) and hence does not fit with our introduction, so we request not to include.

Removed sentence on Page 2, Line 65:

“However, all these initial proteome-wide systems genetic studies in mouse GRPs have focused on liver owing to its essential role in whole-body metabolism”

Replaced with:

“More recently, several studies have performed proteomic analysis of additional tissues from cohorts of the BXD (PMID: 29945935) and CC/DO (PMID: 36334589) and include further phenotypic associations.”

3) Line 114: the RPL7 cis-pQTL that also almost certainly explains RPL19 and RPL23 trans-pQTLs is nice to see, especially as I see there are no other ribosomal genes in the area. I know this is a protein-focused paper, but (a) do you see the RPL7 cis-pQTL in other tissues you have already analyzed, like liver, and (b) are there cis/trans-eQTLs for any of these specific genes in the HMDP for other datasets you have processed? While by itself this seems rather irrelevant to your paper, what I mean is for the phenotype/cis-pQTL pairs you do in Figure 3, i.e. connecting muscle cis-pQTLs to HOMA indices, it would be nice to have a ballpark estimate of how many cis-pQTLs you would even expect to see across-tissue at the cis-pQTL level. Somewhat less relevantly, it would also be nice to know how many of those are also cis-eQTL or trans-eQTL pairs, although I think that has been fairly extensively studied by now (off the top of my head trans-pQTL-trans-eQTL pair: very, very unlikely; cis-eQTL-cis-pQTL pair: maybe 30-50% probability).

Comparing the skeletal muscle pQTL data with previously published datasets (pQTL/eQTL) of the HMDP results in approximately 29-36% overlap with other studies, in line with the reviewer expectations.

Specifically, the cross-tissue overlap between skeletal muscle cis-pQTLs and liver cis-pQTLs is 36.0%. In this comparison, the skeletal muscle pQTL data is the same as used in Supplementary File 2 (this manuscript) and the liver pQTL data is Table S7 from [PMID: 30814737]. The cross-tissue overlap between skeletal muscle pQTLs and liver eQTLs is 29.3% (cis) and 5.1% (trans). The cross-tissue overlap between skeletal muscle pQTLs and heart eQTLs is 33.4% (cis) and 7.5% (trans).

Although we agree these are interesting, we request not to include into the revised manuscript as we are already cautious of the length of text. We feel cross-tissue pQTL and eQTL analysis is not the focus of the article and would detract from the main findings of the paper focused on phenotypic associations and validation of UFMylation as a regulator of muscle function.

4) Line 123: "Among the 527 proteins with a cis-pQTL association, 212 had an intragenic association." How was this determined? It would be nice to have a supplemental table list of all of these proteins and their variant.

These values were determined by analyzing the pQTL data included in https://github.com/JeffreyMolendijk/skeletal_muscle. After running the code up to line 121, we first counted the number of unique protein accessions in the ‘pqtl_m’ object after filtering for all cis-pQTLs with p < 1e-4 (527). Next we counted the number of protein accessions where we only considered cis-PQTLs with p < 1e-4, where the variant is located between the start and end location of the gene (212). A list of skeletal muscle pQTLs with Ensembl Variant Effect Predictions is also available in Supplementary File 2.

5) Lines ~190-215: There are a lot of associations here between genes with cis-pQTLs and phenotypes which I wouldn't really associate so strongly with the muscle proteome in the first place like visceral fat. Of course, if a gene has a cis-pQTL in one tissue it's highly likely (~50% if I remember right) to have a cis-pQTL in any other tissue where it's also expressed. The authors do mention this caveat, e.g. on line 197 mentioned that the pancreas would make more sense to check for insulin concentration phenotypes. There's also the concern that in Figure 3B that we're looking at linear correlations here, and correlating any protein with something like plasma insulin or HOMA-IR can dramatically change depending on a million different variables. For instance, were these plasma insulin calculated in exactly the same individuals as in this muscle proteome study or just the same strains? If just the same strains are the fasted states performed in exactly the same way? I know everyone does this – and me too – I just wonder how reasonable it is. The other examples shown here that you actually follow up on later in the paper, like grip strength and lean mass make sense. One question here though: grip strength also tends to be a rather unreliable measurement if the data for the HMDP strains were not all performed by exactly the same technician and in the exact same mice as measured here. Are all these phenotypes from the same exact females from this study? Or just the same strains? On that note, what's the observed heritability for all of these traits? That might be nice and easy to have on coffeeprot. Of course, this whole figure is "just" hypothesis selection so the selection criteria don't need to be bulletproof (even if it is certainly better if they are), but for instance, I couldn't figure out if the phenotypes were from exactly the same mice as the proteome measurements.

All protein and phenotype measurements were performed on fasted mice and include the same strains, but not on the exact same mice. Indeed, the novel skeletal muscle proteomics data is associated with previously acquired traits from various HMDP cohorts by averaging the values of biological replicates in each strain. Our groups and others have previously demonstrated that, with sufficient sample size in terms of strain number, these averages can be integrated across different mice to define new molecular mechanisms of biology, including fatty liver disease (PMID: 29361464) inter-organ signaling (ref PMID: 29719227 and 36137043) and sex differences of metabolic traits (ref PMID: 30639359). A summary of the studies and citations used in this resource are shown in Supplementary File 4. All grip strength assessment was performed by a single investigator (Peixiang Zhang). As pointed out by the reviewer, correlating, or associating a large number of proteins and phenotypes is expected to discover many interactions that are not easily explained. Furthermore, the associations performed across cohorts may introduce additional variation resulting in false negatives and/or postives. In our case, we try to discover independent data sources that agree with our findings, prior to validating a target in further experiments. For example, UFC1 has associations with hand grip strength in the UK BioBank data, and EPHX1 has associations with Type 2 diabetes (Figure 3). One important consideration that the review eludes to is the reliability in penetrance of genetic associations being dependent on overall heritability of traits. An additional statement has been added to the discussion to highlight this point to the reader. Further, we have added additional comments throughout the manuscript to highlight our proteomic analysis was performed on separate mice to the phenotypic data:

Abstract:

“…proteomic analysis of gastrocnemius muscle from 73 genetically distinct inbred mouse strains, and integrated the data with previously acquired genomics and >300 molecular/phenotypic traits via….

Inserted Results on page 2, line 84:

“The proteomic data were integrated with previously acquired genomic and various molecular/phenotypic data via systems genetics analysis (Figure 1A).”

Inserted Results on page 4, line 145:

“Note that data integration was performed at the strain-level, since the proteomic data was not generated from the same mice as those used in previous studies.”

Limitations of Study”on page 8, line 335:

“We only performed proteomics on female mice from the HMDP, whereas phenotypic analyses were performed on several separate cohorts using both sexes.”

Limitations of Study”on page 8, line 346:

“When interpreting genetic associations, it is also relevant to consider the overall heritability of a given trait. Specifically, broad sense heritability measures can inform the overall confidence in linking genotype to phenotype and inferring genetic interactions with environment and sex {Andreux2012, Seldin2019, Ashbrook2021}. The genetic repeatability (R) for each trait, as determined using the rptR workflow, is reported in Supplementary file 4 {Stoffel2017}. For traits which exhibit a high degree of technical variability such as cardiac function or grip strength, these estimates provide a quantitative metric with which to guide genetic contributions.”

(6) Which HMDP strains have the SNPs that cause the QTL in UFC1? Is it relatively common in the HMDP or only a few strains? If relatively common, e.g. if it segregates between B6 and D2 or between any of the CC founders, is the same effect seen more widely? If it is common in any strain used for mouse population studies, it would be feasible to indicate whether the mechanism here found in females is more general.

The SNPs that are associated with UFC1 protein abundance are very common, almost affecting half of the cohort tested in this study. We visualized the 100 UFC1 cis-pQTLs with the lowest p-values, and noticed genetic separation of mice with low and high UFC1 protein abundance (Author response image 1). The reviewer is correct that the SNPs underlying UFC1 cis-pQTLs segregate the DBA/2J and C57BL/6J mice, founding strains of the BXD cross. Strains containing the DBA/2J alleles have lower levels of UFC1 while strains containing the C57BL/6J allele have higher levels of UFC1. Furthermore, phylogenetic analysis of 38 inbred mouse strains [PMID: 30589851] revealed the loci associated to UFC1 abundance are similar between closely related strains. For example, C57BL/6J and C58/J have similar variants with higher levels of UFC1 abundance, whereas DBA/2J resembles A/J, CBA/J, PL/J, FVB/NJ, SJL/J and BALB/cByJ with lower levels of UFC1. These data imply the pQTL associations are more general and are not specific to females. Interestingly, we did not identify significant UFC1 cis-pQTLs in the liver from our previous analysis of the HMDP despite quantification in >116 mice (57 strains) and 29% sequence coverage (7 peptides) suggesting the observed association may be enriched in skeletal muscle.

Although we agree these observations are of interest and we thank the reviewer for prompting us to perform these analysis, we would prefer not to include these data into the manuscript. We feel it is a little out of place to perform the analysis on only UFC1 (and not other targets with phenotypic associations) and we are already concerned at the length of text. We do not feel the inclusion of these data significantly improve the outcomes of the manuscript.

**Author response image 1. sa2fig1:** Comparison of UFC1 abundance and cis-pQTL SNPs in skeletal muscle of several inbred mouse strains.

7) Figure 5 in general: This looks convincing. I would also imagine, even though it's not my field, that there are now many proteomics studies of muscle proteome expression in ALS mice, as well as mice with other skeletomuscular disorders (DMD?). It would be interesting to know if UFC1 – and the other targets as well – are differentially regulated in such cohorts, especially as the phenotypic effect size here with the shRNA is huge. What about humans with ALS? Checking muscle proteomes in humans and observing differences in UFC1 (or really any of these gene targets) would be a huge finding, and it should not be so hard to check this through collaborators with access to human muscle datasets for those patients with ALS, DMD, or what have you.

We thank the reviewer for this suggestion and have indeed identified previous transcriptomics studies showing an association between changes in the expression of UFMylation genes in other settings of muscular atrophy.

We have updated this in our discussion on page 7, line 316:

“Furthermore, the expression of UFC1 is also up-regulated in skeletal myocytes differentiated from induced pluripotent stem cells (iPSCs) derived from familial ALS (C9ORF72 mutations) [PMID: 34310943], and the expression of UBA5, the E1 ligase for UFMylation has recently been associated with Becker muscular dystrophy [PMID: 33883925].”

8) In a related question, so you have run proteomics on the EDL muscles of a few mice; do the other proteins identified in Figure 3 also change, i.e. are any of them among the top 573? The findings highlighted here are also particularly interesting to me as I doubt any of them would be visible at the mRNA level as they all are proteins involved in large complexes (SEC61, RPL, TRAP, …), which could well indicate why this mechanism has been missed before in literature despite it being a highly-studied phenotype. Conversely, since the theorized mechanism is for hitting histone H4 and a bunch of other machinery involved in both transcription and translation, I would expect to see some signal at the mRNA level, even if it's not the same set of genes at all as at the protein level.

Comparing the targets shown in Figure 3 (located at the Qrr1 region) with the shUFC1 experiment indicated that MPZ, EPHX1, PCP4L1, DUSP23 and NIT1 were not altered. Only the abundance of BPNT1 significantly decreased following UFC1 knockdown. Unfortunately, no RNA sequencing was performed on these same samples. Histone H4 has previously been shown to by UFMylated but we currently do not know if this site is regulated during models of muscle atrophy.

9) I don't see the fasted state mentioned in any protocol except for the HMDP. For instance, the protocol for the ALS animals has the Hilton 2017 paper cited for "as previously described" but that paper doesn't appear to mention fasting. Same for the AAV study, and since this was done in Australia and the HMDP in UCLA, I imagine there are some differences, if not in fasting then perhaps in other criteria. This is only a tidbit here since the actual validation of the UFC1 target clearly worked, but it could be a much larger issue if the authors had, say, followed up on the plasma insulin phenotype highlighted in Figure 3.

The reviewer raises an excellent point. All mice from the HMDP were analysed in the fasted state while all ALS mice and those receiving rAAV6 were analysed in the non-fasted state. We completely agree that this would be a major issue following investigations on metabolic states such as glucose/insulin interventions. As a side note and based on unpublished data, we see very few differences in the mouse skeletal muscle proteome in the fed or fasted state even with deep proteome coverage (~7K proteins). This is in stark contrast to large changes in the fed vs fasted liver proteome [PMID: 32160557]. We have updated the methods to indicate the non-fasted state of the ALS- and AAV-treated mice.

10) This is not a protocol paper so I am not judging it off of coffeeprot, but I have some issues with the website. For instance, I wanted to download the grip strength data and calculate heritability myself, but when I download the data, it just has an average value per strain, no mention of n and SEM, nor all values per strain – just a simple vector. For correlations and mapping, it is very easy to use. I think I already asked this – but are the phenotypes listed here from exactly the same HMDP mice used for this study? Certainly, the utility of the HMDP is that they do not need to be, but if not I am wondering then for traits like "Liver Weight_fast" if this is males, females, mixed, …

We performed strain-level analyses in MuscleProt since the proteomic data acquired in this study was generated from different mice, than those used to generate the phenotypic data. As such, we were unable to compare protein and phenotypic data at the individual mouse level. As mentioned above, we have edited the text in several places to make this more clear.

Abstract:

“…proteomic analysis of gastrocnemius muscle from 73 genetically distinct inbred mouse strains, and integrated the data with previously acquired genomics and >300 molecular/phenotypic traits via….

Results on page 2, line 84:

“The proteomic data were integrated with previously acquired genomic and various molecular/phenotypic data via systems genetics analysis (Figure 1A).”

Results on page 4, line 145:

“Note that data integration was performed at the strain-level, since the proteomic data was not generated from the same mice as those used in previous studies.”

“Limitations of Study” on page 8, line 335:

“We only performed proteomics on female mice from the HMDP, whereas phenotypic analyses were performed on several separate cohorts using both sexes.”

11) I am a bit confused about Supplemental Table 1. It is great this is available, but it doesn't seem to match exactly the protein table I get from Coffeeprot. Also with the table downloaded from coffeeprot, I see it goes from -3.866 to +2.695 across 4027 proteins and 33 strains, but in Table S1, I get values from -6.78 to +6.35 across 5350 proteins from 161 cohorts. OK, I see that it appears to just be removing the strain and protein line if the phenotype was not measured in that strain, and removing proteins if they are NA in all of the strains that do have the phenotype measured. The confusion comes when I look at something that is paired; for instance, the expression of E9Q1W3. In Table S1, I get values of 0.13 and 0.07 for AXB10 (#51 and #52) but in the CoffeeProt file, I get just a single value of 0.154 for AXB10, which is not clearly related to the 0.13 and 0.07. There is probably an easy explanation here, but I'm not getting it. On a related note, the units for the figures are a bit confusing, e.g. Figure 1F has no units for the proteins, but FigureI1 has the units as log2AUC. Aren't they all ratios based on the TMT control? Or have additional transformations been carried out?

Supplemental Data 1 contains the proteomic data exported from Proteome Discoverer for each mouse (normalized to the internal standard of each TMT batch). The data available in MuscleProt further normalizes these data (quantile normalization) followed by averaging each biological replicates in each strain. These is done to allow associations to previously acquired phenotypic data as described above. We have decided to include the raw exported data from Proteome Discoverer as Supplemental Data 1 as we have previously been asked to provide these data in previous manuscripts.

We included further information in the methods on page 11, line 508:

“For OPLS and protein-trait correlations, proteomic data was quantile normalised and the biological replicates within each strain averaged.”